# Lifestyle mediates the role of nutrient-sensing pathways in cognitive aging: cellular and epidemiological evidence

Chiara de Lucia [1], Tytus Murphy[1], Claire J. Steves[2], Richard J.B. Dobson [3], Petroula Proitsi [1] & Sandrine Thuret [1,4]✉

Aging induces cellular and molecular changes including modification of stem cell pools. In particular, alterations in aging neural stem cells (NSCs) are linked to age-related cognitive decline which can be modulated by lifestyle. Nutrient-sensing pathways provide a molecular basis for the link between lifestyle and cognitive decline. Adopting a back-translation strategy using stem cell biology to inform epidemiological analyses, here we show associations between cellular readouts of NSC maintenance and expression levels of nutrient-sensing genes following NSC exposure to aging human serum as well as morphological and gene expression alterations following repeated passaging. Epidemiological analyses on the identified genes showed associations between polymorphisms in SIRT1 and ABTB1 and cognitive performance as well as interactions between SIRT1 genotype and physical activity and between GRB10 genotype and adherence to a Mediterranean diet. Our study contributes to the understanding of neural stem cell molecular mechanisms underlying human cognitive aging and hints at lifestyle modifiable factors.

[1] Department of Basic and Clinical Neuroscience, Institute of Psychiatry, Psychology and Neuroscience, King's College London, London, UK. [2] Department of Twin Research and Genetic Epidemiology, King's College London, London, UK. [3] Department of Biostatistics and Health Informatics, Institute of Psychiatry Psychology and Neuroscience, King's College London, London, UK. [4] Department of Neurology, University Hospital Carl Gustav Carus, Technische Universität Dresden, Dresden, Germany. ✉email: Sandrine.1.thuret@kcl.ac.uk

Ageing can be defined as the time-dependent functional decline that affects most living organisms. At a cellular level, it is accompanied by several hallmarks including genomic instability, oxidative stress, cellular senescence, deregulation of nutrient-sensing pathways, senescence and stem cell exhaustion[1].

Human ageing, in particular, is characterised by vast heterogeneity at both a cellular and organismal level[2]. This heterogeneity has given rise to the notion that biological and chronological age differ from one another; the former being the advancement of the biological ageing process and the latter being the passing of time[3].

A typical ageing phenotype with clear discrepancy between biological and chronological age is cognitive decline; while some individuals retain intact cognitive performance, others display dramatic deterioration[4]. Brain ageing presents with vast alterations throughout several sub-regions, each playing an important role in determining its phenotypic presentation both in ordinary and pathological conditions[5]. Studies aimed at identifying mechanisms responsible for age-related cognitive decline and Alzheimer's dementia, for example, have emphasised an important role for the hippocampus and its residing neural stem cells (NSCs)[6]. In humans, episodic memory is the cognitive function mapped to the hippocampus and it is particularly important in age-related cognitive decline and progression to dementia[7,8]. In line with this, rodent models have repeatedly linked NSC maintenance and the resulting adult hippocampal neurogenesis to ageing and cognitive decline[9–11]. Human adult neurogenesis has been debated for decades and recently prominent journals have published opposing results about its presence and rate of decline during ageing[12,13]. Yet, the general consensus remains that a modest but important neuronal turnover occurs in the human neurogenic niche and that this process is likely involved in cognitive ageing and neurodegenerative disorders[14–17].

Interestingly, studies investigating NSC function in cognitive ageing have shown that the circulatory system alone can affect NSC maintenance, neurogenesis and cognitive decline[11,18]. This suggests that environmental factors able to alter the composition of the systemic environment, such as diet and exercise, can affect the rate of age-related cognitive decline potentially explaining some of ageing's heterogeneity[19]. Indeed, recent longevity studies suggest only a minor role for genetics and a far greater effect of environmental factors than initially predicted[20–22]. While calorie restriction is the most widely studied dietary intervention with respect to longevity, diets based on the presence of specific nutrients have been repeatedly linked to improved cognition and neurogenesis during ageing[23–25]. Similarly, exercise has been associated to improved cognitive abilities in both animal and human studies, suggesting lifestyle can impact the rate of cognitive decline[26–28].

Animal models have shown nutrient-sensing pathways, such as the insulin and insulin-like growth factor1 (IIS), mTOR and sirtuin pathways, can provide the molecular basis for the association between lifestyle and ageing. Furthermore, these pathways have been implicated in stem cell maintenance, suggesting they could also be involved in the interaction between lifestyle, NSCs and cognition[29]. Though animal models have provided overwhelming evidence supporting a role for the mTOR[30], sirtuin[31] and IIS pathways[32] in longevity, only limited studies have confirmed their role in human models[33]. Notably, among these are several studies suggesting a link between FOXO3a polymorphisms and human longevity[21,33,34]. Identifying the driving mechanisms behind cognitive decline and understanding the environmental and genetic causes of its phenotypic heterogeneity will provide new therapeutic avenues.

The present study aims to assess the contribution of nutrient-sensing pathway to human NSC and cognitive ageing starting from in vitro molecular and cellular findings back-translated to epidemiology and genetic data (Fig. 1). To this aim, we focus on 16 candidate genes belonging to the mTOR, sirtuin and IIS nutrient-sensing pathways. Key genes in each pathway were selected via literature searches (summarised in ref. [33]) based on both previous studies confirming roles in ageing and on their role in the signalling cascade; genes spanning from receptors to transcription factors were selected to allow for a deeper understanding of potential underlying mechanisms.

We first assessed the effect of the ageing systemic environment on the expression of candidate genes and then selected those showing associations to NSC maintenance markers for further analysis using a multimodal in vitro model of NSC ageing (Fig. 1a). Finally, given the contribution of both genetic and environmental factors to cognitive decline, we assessed both their individual and combined effect on a hippocampus and age-dependent cognitive task in a cohort of 1638 twins (Fig. 1b). Polymorphisms in the selected candidate genes, physical activity and three dietary measures: adherence to Mediterranean diet (MDS), healthy eating (HEI) and daily calorie intake were thus tested for association to Paired Associates Learning (PAL) performance. We report interesting alterations in the expression levels of FOXO3A, NAMPT, PTEN and GRB10 following the in vitro models and significant associations between single nucleotide polymorphisms (SNPs) in ABTB1 and SIRT1 with cognitive performance. Finally, we show interactions between lifestyle, cognitive performance and polymorphisms in SIRT1 and in GRB10.

## Results

**Nutrient-sensing pathways candidate gene expression is associated to cellular readouts.** To assess the role of the ageing systemic environment in a human context, we used an in vitro parabiosis model during which human NSC were treated with serum from young or old individuals.

The expression levels of 16 candidate nutrient-sensing genes showed no age-dependent variation but showed considerable variation in response to treatment with different serum samples (Supplementary Fig. 2). We next assessed whether this variation was associated to cellular readouts of stem cell maintenance. Therefore, mRNA levels of each candidate gene were tested for association to cell density to infer overall cell health and to markers of stemness, proliferation, differentiation and apoptosis. Pearson's correlations revealed several significant associations.

Within the IIS pathway, expression of the FOXO3A transcription factor correlated with detected apoptosis levels (CC3) ($r = -0.42$, $p = 0.022$), the number of immature neurons (MAP2) ($r = 0.48$, $p = 0.009$), and cell density ($r = 0.76$, $p < 0.0001$) showing increased FOXO3A expression was associated to increased numbers of cells in general and immature neurons in particular (Fig. 2a–c). In line with this, expression of PTEN (Fig. 2d), a negative regulator of the IIS pathway, and of IGF2R (Fig. 2e), the IIS pathway's scavenger receptor, showed significant negative associations to cell density ($r = -0.41$, $p = 0.031$ and $r = -0.53$, $p = 0.003$ respectively).

Within the mTOR pathway, expression of mTOR ($r = 0.42$, $p = 0.016$) (Fig. 2f) and of its negative regulator, GRB10 ($r = 0.45$, $p = 0.012$) (Fig. 2g), positively correlated to cell density. Importantly, GRB10 expression was also significantly associated to both mTOR and PTEN expression (Supplementary Fig. 2QR).

Additionally, expression levels of NAMPT ($r = 0.47$, $p = 0.012$) (Fig. 2h) and UCP2 ($r = 0.48$, $p = 0.008$) (Fig. 2i), in the sirtuin pathway, showed positive associations to proliferation (KI67).

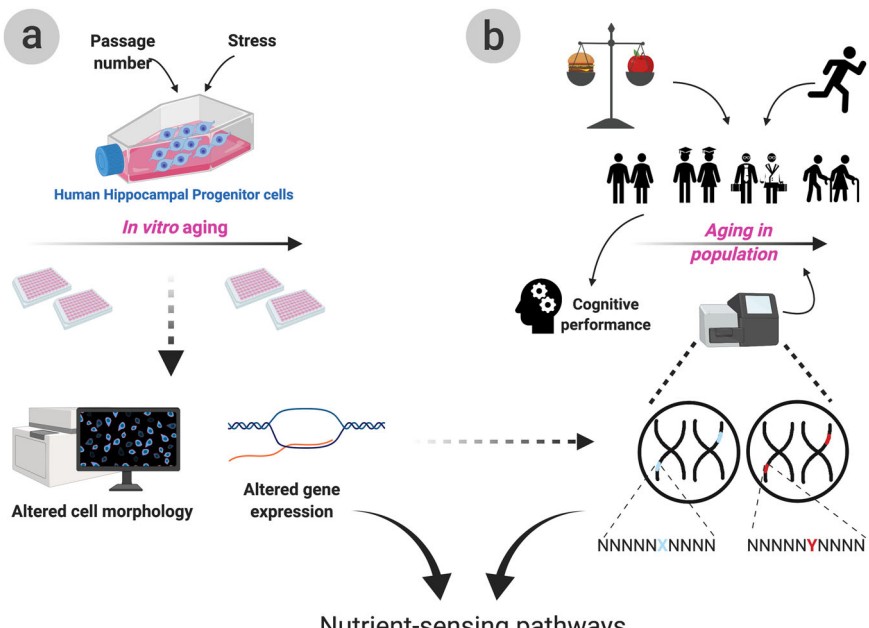

**Fig. 1 Graphical summary of the cellular and epidemiological experiments.** The ageing process was mimicked in vitro via the treatment with HU (hydroxyurea), tBHP (tert-butyl hydroperoxide) and repeated passaging of human hippocampal progenitor cells (HPC). The HPCs were assessed for cellular alterations via morphological and immunohistochemical analysis and for molecular alterations via qPCRs (**a**). We report morphological alterations reminiscent of a senescent phenotype as well as several gene expression alterations in nutrient-sensing pathway genes following in vitro ageing. In parallel, epidemiological techniques were employed to assess how diet, exercise and genetics interact to affect age-related cognitive performance throughout ageing (**b**). Dietary measures such as Healthy Eating Index (HEI), Mediterranean Diet Score (MDS) and daily calorie intake, as well as results from the International Physical Activity Questionnaire (IPAQ) were employed to assess the role of lifestyle on cognition. The role of genetics was also assessed by investigating the association between polymorphisms in nutrient-sensing genes identified in (**a**) and cognitive performance. We demonstrate an important contribution of both lifestyle and genetics to age-dependent cognitive performance.

Finally, increased ABTB1 expression correlated with increased cell number (cells per field) ($r = 0.58$, $p = 0.001$), an increase in immature neurons (MAP2) ($r = 0.70$, $p < 0.0001$) and a decrease in apoptotic cell death (CC3) ($r = -0.69$, $p < 0.0001$) and proliferation ($r = -0.43$, $p = 0.024$) (Fig. 2j–m).

Though expression levels of SIRT1, MASH, ETV6, S6K, EIF4E, 4EBP1, IRS2 and NRIP showed no correlation to cellular readouts, the associations described above support an important role for nutrient-sensing pathways in stem cell maintenance. mTOR, GRB10, FOXO3a, PTEN, IGF2R, SIRT1, NAMPT, UCP2 and ABTB1 were therefore selected for further analysis. SIRT1was included despite showing no correlation to cellular readouts, given the promising results of both its downstream target (FOXO3A) and upstream regulator (NAMPT).

**Further passaging results in morphological alterations.** Given the heterogeneity of human serum, a more controlled model was established to further assess the role of these genes in NSC ageing. In this alternative in vitro ageing model, stress was induced by pharmacological treatment of the NSC with tert-butyl hydroperoxide (tBHP), to induce oxidative stress, and hydroxyurea (HU), to induce replication stress, combined with increased passage number, mimicking older cells.

Cell count based on DAPI (4′,6-diamidino-2-phenylindole) stain was used to determine viable concentrations of tBHP and HU while markers of proliferation and differentiation were used to assure the NSC were able to proliferate and differentiate at higher passage numbers (Supplementary Fig. 3).

Following pilot experiments showing negligible differences between cells with passage numbers higher than 26 (Supplementary Fig. 4), passage 26 was selected as the 'old' passage number

for experimental ease. Passage 17 was selected as the 'young' passage as it was the minimum number of passages enabling sufficient yield to carry out both cellular and molecular work.

To test whether in vitro ageing affected NSC maintenance, we assessed cellular and morphological measures following treatment with TBHP and HU and increased passage number. Though cellular readouts remained largely unaltered (Supplementary Fig. 5) and there were no alterations in number of neuroblasts or immature neurons, there were noticeable morphological differences in these cell types.

In cultures with higher passage numbers, both DCX (neuroblasts) and MAP (immature neurons) positive cells appeared rounder than their lower passage counterparts. Machine learning was employed to classify cells into two classes, characterised by either a stunted or elongated morphology.

Two-way ANOVAs of machine learning results regarding MAP2-positive cells showed significant variation between groups and that 83% of the variation was due to passage number ($F(1,8) = 41.39$, $p = 0.0002$). There was no effect of treatment or interaction effect but post-hoc analysis revealed that in both the untreated ($t(8) = 5.096$, $p = 0.0009$) and treated groups ($t(8) = 4.55$, $p = 0.004$) (Fig. 3a), this variation was due to an increase in the proportion of MAP2-positive cells with stunted morphology in the high-passage group when compared to their proportion in the low-passage group.

Similarly, two-way ANOVAs of machine learning results on DCX-positive cells showed significant variation between groups and that 81% of the variation was due to passage number ($F(1,8) = 35.28$, $p = 0.0003$). As for MAP2 cells, there was no effect of treatment or interaction effect. In both the untreated ($t(8) = 4.503$, $p = 0.002$) and treated groups ($t(8) = 3.90$, $p = 0.005$) (Fig. 3b), there was an increase in the proportion of DCX-positive

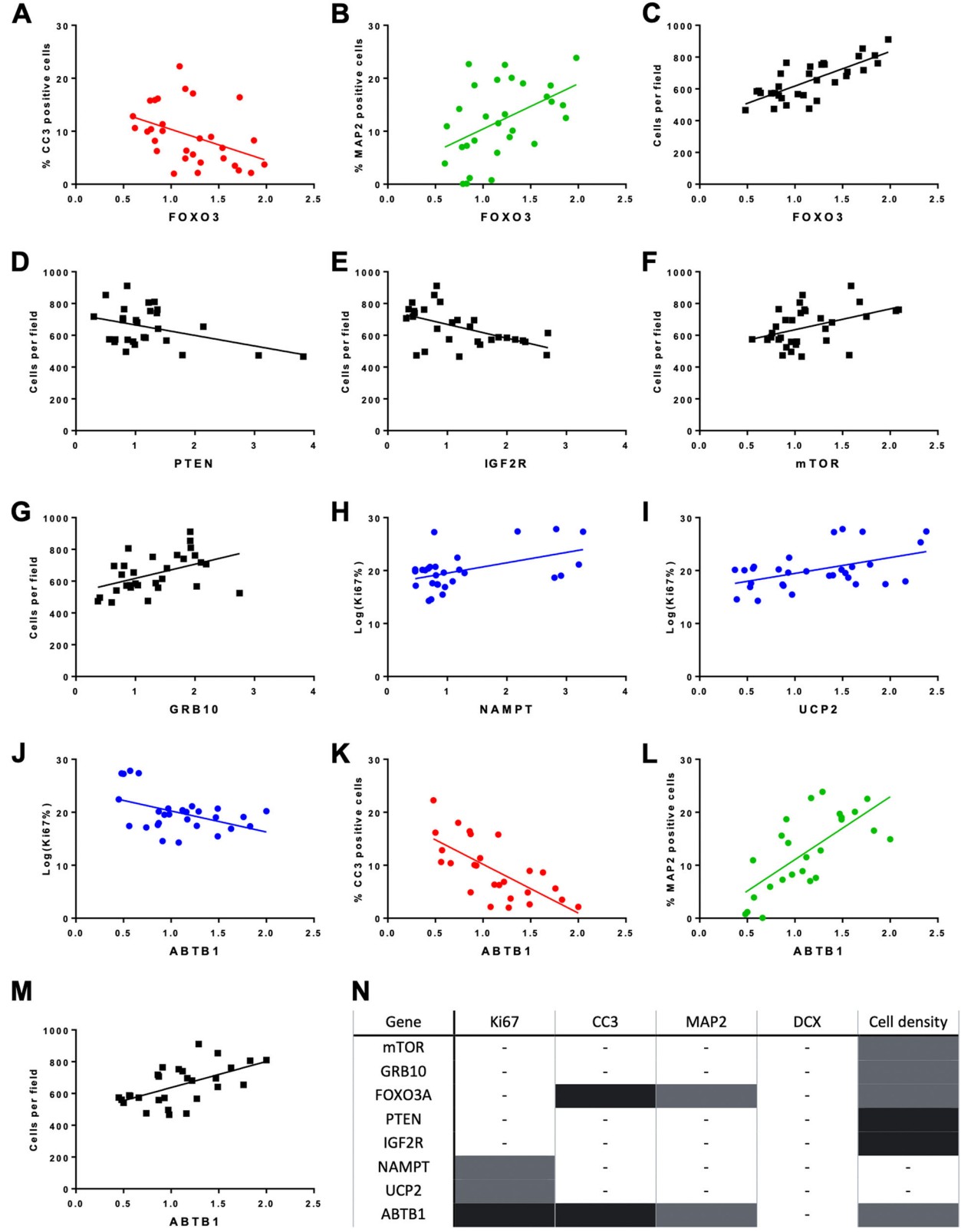

cells with stunted morphology in the high-passage group when compared to their proportion in the low-passage group. For both MAP2- and DCX-based machine learning results, Fisher's LSD post-hoc analysis was carried out followed by Bonferroni correction.

Importantly, machine learning results showed that similar factors were used to determine whether each cell belonged to the stunted or the elongated morphology subgroup for both MAP2- and DCX-positive cells. Cytoplasm width to length ratio, nucleus area and nucleus roundness were used by both the MAP2- and DCX-based paradigms. In addition, cytoplasm width was used in the MAP2-based machine learning while cytoplasm roundness was used in the DCX-based one.

**Fig. 2 Expression levels of candidate genes in nutrient-sensing pathways are associated to cellular readouts following in vitro parabiosis assay. a–m** Scatterplots showing the significant correlations between markers of cell death (CC3), immature neurons (MAP2), proliferation (Ki67) and cell number on the y-axis and gene expression of candidate genes on the x-axis. The data reported here shows markers of stem cell maintenance are associated to the expression levels of 8 out of 16 candidate genes analysed. Each dot represents a participant and the line of best fit is shown in each graph. Correlations with CC3 are in red, with MAP2 in green, cell number in black and Ki67 in blue. Gene expression relative to control as calculated by the Pfaffl method. Normality was tested using the Shapiro−Wilks normality test. All Ki67 datasets did not show a normal distribution and were log transformed to ensure a normal distribution. **n** Table summarising the associations between cellular marker and gene expression displayed in the scatterplots (**a**−**m**). Grey indicates a positive association, black indicates a negative association.

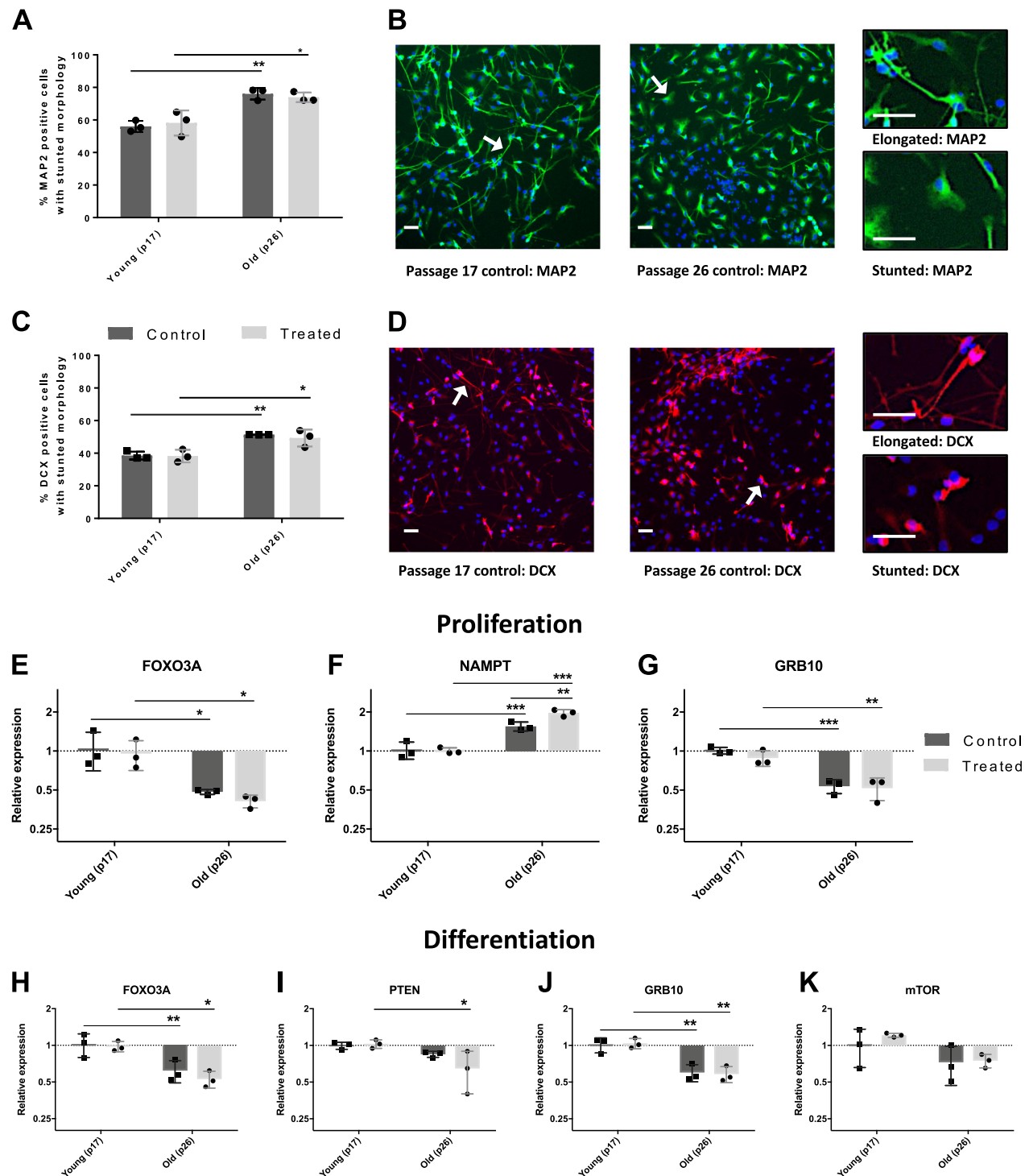

**Fig. 3 Increased passage number is accompanied by morphological and gene expression alterations. a** Graph showing the machine learning results assessing the percentage of MAP2-positive cells with stunted morphologies. Control and treated passage number 17 (p17) and passage number 26 (p26) cells were compared to one another. The results show an increase in the percentage of cells with a stunted morphology in p26 cells when compared to p17 and no alterations due to pharmacological treatment. Each dot represents the average result of three technical replicates. **b** Graph showing machine learning results for the percentage of DCX-positive cells with stunted morphologies in passage number 17 (p17) and passage number 26 (p26) cells. As for MAP2-positive cells, results show an increase in the percentage of DCX-positive cells with a stunted morphology in p26 cells when compared to p17 and no alterations due to pharmacological treatment. Each dot represents the average result of three technical replicates. **c, d** Representative images for each marker are reported on the right. DAPI (blue), MAP2 (green), DCX (red). Scale bar: 50 μm. **e–k** Graphs showing the expression levels of candidate genes in control and treated p17 and p26 cells relative to control (p17 control cells) as per Pfaffl method. Each dot represents the average result of three technical replicates. FOXO3A (**e**), NAMPT (**f**) and GRB10 (**g**) expression levels showed alterations due to passage number following proliferation assay. NAMPT also showed a cumulative effect of pharmacological treatment and increased passage number as evidenced by the significant increase in expression in treated p26 cells when compared to treated p17 cells. FOXO3A (**h**), PTEN (**i**) and GRB10 (**j**) and mTOR (**k**) expression levels showed alterations due to passage number following differentiation assay. The variation in mTOR expression due to passage number did not survive Bonferroni correction. Control: media-only conditions. Treated conditions indicate cells treated with 0.01 μM tert-butyl hydroperoxide (tBHP) and 10 μM hydroxyurea (HU). Bar graphs denote mean ± SD. Two-way ANOVAs with Bonferroni correction, three biologically independent experiments. *$p < 0.05$; **$p < 0.01$; ***$p < 0.001$. For graphs (**c–i**) y-axis are logged for ease of visualisation.

---

**Further passaging alters FOXO3A, NAMPT, PTEN and GRB10 expression.** Having assessed the cellular changes, we focused on the expression levels of the nine candidate genes highlighted by the in vitro parabiosis model.

Though treatment with tBHP and HU left candidate gene expression unaltered (Supplementary Fig. 6), passage number caused interesting variations. The effect of passage number on expression levels was investigated as well as the effect of combining tBHP and HU treatment with passage number (Fig. 3e–k). Following the proliferation assay, two-way ANOVAs highlighted significant variation in FOXO3A, NAMPT and GRB10 expression levels.

Analysis of FOXO3A expression showed 70% of the variation was due to passage number ($F_{(1,8)} = 20.15$, $p = 0.002$), while there was no effect of treatment or interaction. Accordingly, post-hoc analysis revealed significant alterations in FOXO3A mRNA levels between the young and old untreated subgroups ($t_{(8)}3.23$, $p = 0.012$) and between the young and old treated subgroups ($t_{(8)} = 3.12$, $p = 0.014$), showing that in both treated and untreated cells, higher passage number causes decreased FOXO3A expression (Fig. 3c).

NAMPT mRNA levels showed 82% of the variation between groups was due to passage ($F_{(1,8)} = 120.2$, $p < 0.0001$), 6% due to treatment ($F_{(1,8)} = 8.73$, $p = 0.018$) and 7% due to interaction effects ($F_{(1,8)} = 10.1$, $p = 0.013$). Post-hoc analysis revealed young and old untreated cells were significantly different ($t_{(8)} 5.50$, $p = 0.0006$) as well as young and old treated cells ($t_{(8)}9.998$, $p < 0.0001$) and old treated and untreated cells ($t_{(8)}4.34$, $p = 0.0025$). Showing passage number alone causes increased NAMPT levels, that there is a summative effect of passage number and treatment, but that treatment alone does not elicit expression changes (Fig. 3f).

GRB10 expression levels analysis showed 85% of variation was due to passage number ($F_{(1,8)} = 63.25$, $p < 0.0001$) and that treatment and interaction had no effect. Post-hoc analysis showed young and old untreated cells displayed significantly different expression levels of GRB10 ($t_{(8)}6.33$, $p = 0.0002$) as did young and old treated cells ($t_{(8)} = 4.92$, $p = 0.0012$) (Fig. 3g). All other genes showed no mRNA level changes between groups following the proliferation assay (Supplementary Fig. 7A).

mRNA levels of the nine candidate genes were also assessed following the differentiation assay. Two-way ANOVAs showed significant variation in the gene expression of FOXO3A, PTEN, GRB10 and mTOR.

75% of variation for FOXO3A was due to passage number ($F_{(1,8)} = 26.3$, $p = 0.0009$) while treatment and interaction showed no significant effects. Post-hoc analysis revealed young

and old untreated cells displayed significant variation ($t_{(8)} = 3.35$, $p = 0.0095$) as did young and old treated cells ($t_{(8)} = 3.84$, $p = 0.005$). Similarly to the proliferation data, increased passage number caused a decrease in FOXO3A expression (Fig. 3h).

PTEN mRNA levels showed 50% of the variation due to passage number ($F_{(1,8)} = 11.37$, $p = 0.0098$) while treatment and interactions showed no effect on variance. Young and old treated cells displayed significantly different PTEN levels ($t_{(8)} = 3.415$, $p = 0.0092$). This was not the case for young and old untreated cells that showed no difference in PTEN expression ($t_{(8)} = 1.354$, $p = 0.21$). These results suggest that treatment of higher passage number cells, but not lower passage number cells, causes a decrease in PTEN levels (Fig. 3i).

Analysis of GRB10 mRNA levels showed 84% of the variance was due to passage number ($F_{(1,8)} = 47.84$, $p = 0.0001$). Treatment and interaction showed no significant effects. GRB10 expression was significantly different between both young and old untreated cells ($t_{(8)} = 4.63$, $p = 0.0017$) and between young and old treated cells ($t_{(8)} = 5.152$, $p = 0.0009$). These results match those of the proliferation assay suggesting GRB10 levels are consistently lowered during both proliferation and differentiation as a result of increased passage number (Fig. 3j).

Finally, mTOR level analysis showed significant variation between groups with 48% of variation attributable to passage number ($F_{(1,8)} = 8.30$, $p = 0.02$) and no significant effect on variation of treatment and interaction effects. There was significant variance in mTOR expression levels between young and old treated cells ($t_{(8)} = 2.54$, $p = 0.035$) but not between young and old untreated cells ($t_{(8)} = 1.539$, $p = 0.16$), suggesting mTOR levels are lowered in cells with higher passage numbers (Fig. 3k). This variation, however, did not survive multiple testing correction. All other genes showed no significant variance following the differentiation assay (Supplementary Fig. 7).

For all analyses involving two-way ANOVAs comparing passage number and tBHP and HU treatment effects, Fisher's LSD following by Bonferroni correction was applied to account for three comparisons. All results reported as significant, except the difference in mTOR mRNA levels between young and old treated cells, survived this threshold.

**Age, NART and education predict cognitive performance.** To assess whether the NSC maintenance-associated genes identified above play a role in cognition, we investigated the association of genetic variation within these genes to performance on a hippocampus-dependent cognitive task in a human population of 1638 twins. As performance on PAL is known to be age and

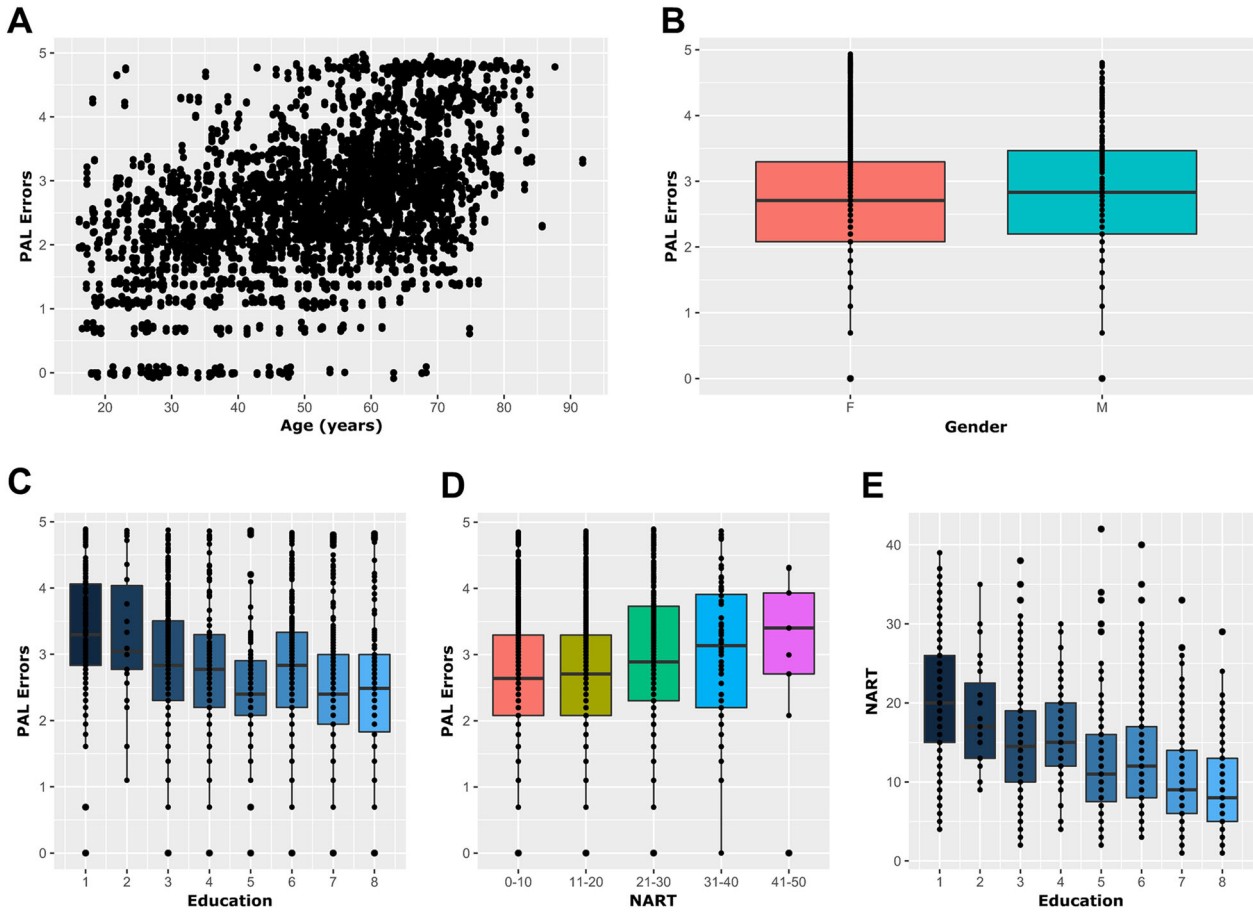

**Fig. 4 Epidemiological analysis in TwinsUK cohort shows age, NART errors and education level are predictive of cognitive performance. a** Scatter plot showing the significant correlation between age and PAL errors suggesting increasing age is associated to a decline in performance. **b** Tukey boxplot showing the non-significant association between gender and PAL error (Welch's *t* test). **c–e** Tukey boxplot showing the significant correlations between education measure and age (**c**), National Adult Reading Scale (NART) and age (**d**) and NART errors and education measure (**e**). These results show that education level measured either via attained qualification or NART performance is an important co-variate when assessing cognitive performance. For (**c–e**), data were treated as continuous but graphed as categorical for ease of visualisation. Each dot represents a participant.

education dependent, we first assessed the role of age, education-level and National Adult Reading Scale (NART) errors on PAL.

A moderate correlation (Pearson's, $r = 0.50$, $p < 0.00001$, $n = 2105$) was observed between age and PAL errors (PALe) (Fig. 4a), while gender showed no effect on PALe (Welch *t* test, $p = 0.1$, $n = 2105$) (Fig. 4b). PALe was tested for correlation with education-level and NART errors. PALe showed significant negative correlation with education (Spearman's, $r = -0.29$, $p < 0.00001$, $n = 1384$) (Fig. 4c). Similarly, NART errors showed a significant correlation with PALe (Pearson's, $r = 0.10$, $p < 0.00001$, $n = 1784$) (Fig. 4d). As NART data were much more complete and accurate than education-level data, and both datasets showed moderate significant negative correlation (Spearman's, $r = -0.43$, $p < 0.00001$, $n = 1185$) (Fig. 4e), NART was selected as the education measure. Age and NART were thus selected as covariates for all models involving PAL. In addition, though gender showed no effect on PAL, it was included as a covariate to account for the possibility of an undetected effect owing to the limited number of male participants within our cohort (1879:226 F:M) and the varying number of participants in each model.

**Physical activity modulates the association between calorie intake and PAL performance.** Next we assessed the association

of lifestyle measures and PAL performance irrespective of geno-type. We tested healthy eating, adherence to Mediterranean diet, calorie intake and physical activity levels as measured by the international physical activity questionnaire (IPAQ)[35,36]. No single lifestyle measure showed significant association to PALe suggesting dietary or exercise habits alone are not predictive of PAL score (Fig. 5a–d). However, there was a significant interaction between physical activity and calorie intake ($\beta = -2.79$, $p = 0.009$) showing different associations between calorie intake and PALe depending on the participant's exercise level (Fig. 5g). For participants with low IPAQ, each Log(Kcal) consumed was predictive of a 2.55 increase in PALe. For participants with moderate IPAQ levels, each Log(Kcal) consumed was predictive of a 0.24 decrease in PALe. Finally, participants with high IPAQ levels showed a decrease of 3.04 PALe for each Log(Kcal) consumed. These results indicate that participants with low physical activity levels show a positive association between increased calories and poorer cognitive performance while those undergoing high physical activity levels show a negative association. Accordingly, those with moderate physical activity levels only showed a modest negative association. There were no significant interactions between physical activity and the remaining diet scores (Fig. 5e) after adjusting for the seven multiple comparisons using the Benjamini–Hochberg (BH) procedure.

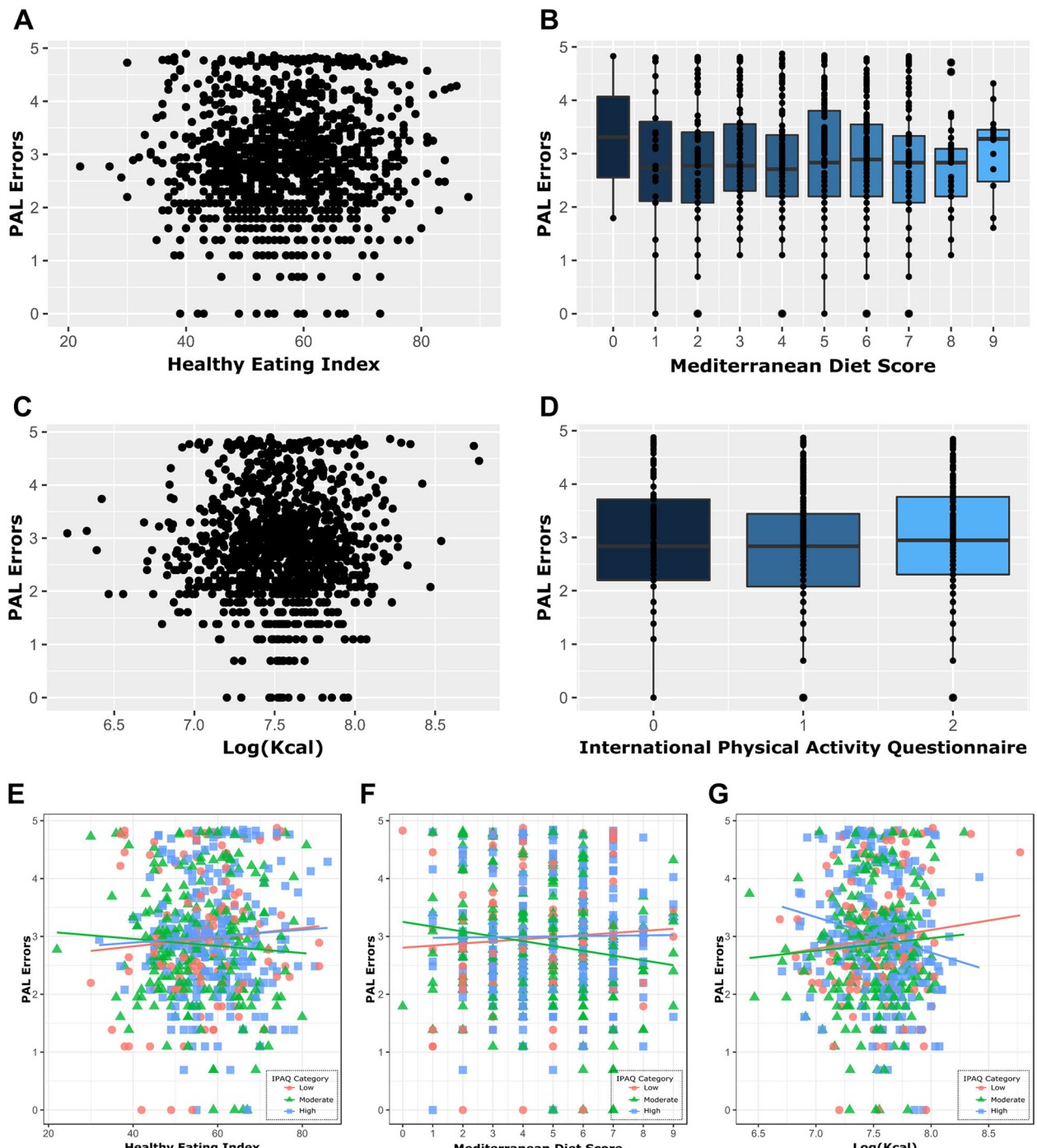

**Fig. 5 Physical activity modulates the association between calorie intake and PAL performance in the TwinsUK cohort.** Individual lifestyle measures showed no association to PAL errors as shown in graphs (**a**–**d**). Scatter plots showing the lack of association between PAL errors and healthy eating (**a**) and between PAL errors and calorie intake (**c**). Tukey boxplots showing the lack of association between PAL errors and adherence to Mediterranean diet (**b**) and between PAL errors and physical activity (**d**). **e**–**g** Scatterplots showing the interaction between physical activity and diet testing whether lifestyle measures can interact and thereby affect each other's association to PAL errors. Physical activity had no effect on the association between healthy eating and PAL errors (**e**) or on the association between adherence to Mediterranean diet and PAL errors (**f**). Physical activity, however, significantly affected the association between Kcal intake on PAL errors (**g**) showing that calorie intake and cognitive performance can have either a positive or negative association depending on the individual's physical activity level.

**SNPs in *SIRT1* and *ABTB1* are predictive of PAL performance.**
Next, generalised estimating equations (GEE) linear models were used to investigate whether SNPs in the nine candidate genes were associated to PALe. Using GEE models, 482 SNPs selected as tag-SNPs were tested for association to PAL performance while including age, gender and NART score as covariates. As the

cohort included twins, GEE clusters were used to account for twin pairing. Significant associations between SNPs rs497849 (an SNP in the proximity of *SIRT1*) ($\beta = 0.19$, $p < 0.001$, $n = 1286$) and rs782431 (an SNP in the proximity of *ABTB1*) ($\beta = 0.14$, $p < 0.001$, $n = 1277$) with PALe were observed (Fig. 6a, b). No other SNPs were in linkage disequilibrium (LD) with rs497849

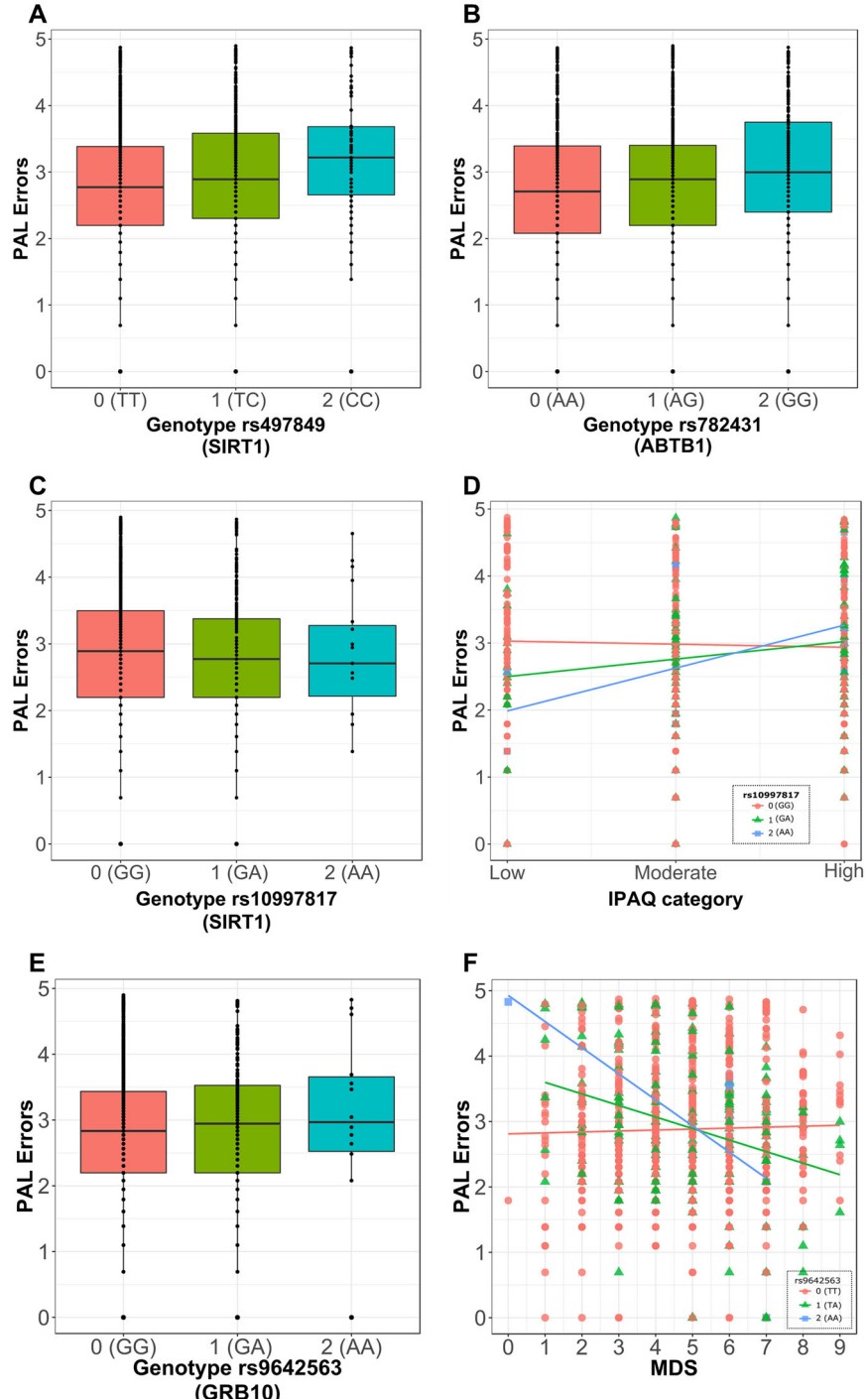

**Fig. 6 GEE models show genotype alone and genotype combined with lifestyle affects PAL performance in TwinsUK cohort. a**, **b** Tukey's boxplots showing significant associations between the rs497849 (**a**) and rs782431 (**b**) genotype with PAL errors. **c** Tukey boxplot showing the lack of association between rs10997817 genotype with PAL errors and **d** scatterplot showing the interaction of rs10997817 genotype on the association between physical activity (IPAQ) and PAL errors. Together, (**c**) and (**d**) show that while there is no significant association between rs10997817 genotype and physical activity, the association between physical activity and PAL errors is modulated by the individual's genotype for the rs10997817 SNP. **e** Tukey boxplot showing the lack of association between rs9642563 genotype and PAL errors and **f** scatterplot showing the significant interaction of rs9642563 genotype on the association between adherence to Mediterranean diet (MDS) and PAL errors. As above, (**e**) and (**f**) show that while there is no significant association between rs9642563 genotype and MDS, the association between MDS and PAL errors is modulated by the individual's genotype for the rs9642563 SNP. Data were analysed as categorical but represented as continuous for ease of visualisation. Genotypes are displayed as 0: homozygous for the major allele (orange), 1: heterozygous (green) and 2: homozygous for the minor allele (blue).

but three SNPs were in LD with rs782431: rs2630213, rs2720262 and rs782455, making them interesting candidates for effects on PAL. Of note, is the presence of a further two tag-SNPs (rs6439080 and rs782448) related to *ABTB1* in the top 5 most significantly associated SNPs to PAL, though these did not survive multiple testing correction (Supplementary Data 1). BH procedure was applied to account for the 482 multiple comparisons needed to test the association of each SNP to PAL performance.

**The association between lifestyle and PAL performance is modulated by SNPs in *SIRT1* and *GRB10*.** Given the interaction between lifestyle and PAL performance, we assessed whether specific genotypes affect the association between lifestyle factors and PAL performance. There were no significant interactions between genotype and healthy eating or calorie intake. However, rs10997817 genotype (an SNP in the proximity of *SIRT1*) mediated the association between IPAQ and PALe ($\beta = 0.30$, $p < 0.001$, $n = 613$), showing varying physical activity levels affect PAL performance differently according to genotype (Fig. 6c, d). For participants who are homozygous for the major allele (genotype 0), an increase in physical activity is predictive of a decrease of 0.054 in PALe. For those who are heterozygous (genotype 1) instead, increasing physical activity units is predictive of an increase of 0.3 PALe. In contrast, participants who are homozygous for the minor allele (genotype 2) show a large increase, of 0.56 PALe, with increased exercise. Importantly, rs10997817 tags rs12220640 and rs77797709, suggesting any of these three SNPs could be responsible for the interaction with physical activity (Supplementary Data 2). Though no other association between SNP, PALe and IPAQ survived multiple testing corrections, three more SNPs (rs866255, rs10997810, rs4457643) in the vicinity of SIRT1 also showed possible interactions between genotype, cognition and physical activity (Supplementary Data 3).

Furthermore, rs9642563 genotype (an SNP in the proximity of *GRB10*) mediated the association between adherence to Mediterranean diet and PAL score ($\beta = -0.17$, $p < 0.001$, $n = 900$), showing diet is differentially associated to cognitive performance based on genotype (Fig. 6e, f). For participants who are homozygous for the major allele (genotype 0), increased adherence to Mediterranean diet is predictive of a 0.02 increase in PALe. For those heterozygous or homozygous for the minor allele (genotypes 1 and 2) instead, adherence to Mediterranean diet is predictive of a 0.14 and 0.31 decrease in PALe respectively. Further supporting this finding is the possible interaction between rs2108350, another tag-SNP in GRB10, and Mediterranean diet adherence which, in this study, did not survive multiple testing correction ($\beta = -0.11$, $p < 0.001$, $n = 943$) (Supplementary Data 4). For each lifestyle measure, BH procedure was applied to account for the 482 multiple comparisons needed to test the interaction of each SNP and that lifestyle measure. BH, however, was not applied to account for the four lifestyle measures tested throughout the study.

Importantly, both genotypes showing interaction with lifestyle measures and PALe showed no association to PALe without lifestyle in the model (Fig. 6d, f).

## Discussion
In this study, we investigated candidate nutrient-sensing genes expression in an in vitro parabiosis model and in a novel model of in vitro ageing. We tested the association between lifestyle factors and cognitive performance in a human cohort, and the role of genetic variation in these candidate genes on cognitive performance. These different approaches converge in highlighting

*ABTB1* and *GRB10* as key genes in the interplay between NSC, ageing, lifestyle and cognition.

This study confirms the involvement of nutrient-sensing pathways in human NSC maintenance and supports their role in NSC ageing. Though the in vitro parabiosis experiments, which mimicked previously published in vivo parabiosis experiments[11,18], showed no effect of serum-donor age on gene expression, they revealed interesting associations between known regulators of nutrient-sensing pathways and cellular markers. These results confirm the systemic environment's ability to alter cellular function, which is of particular interest given evidence suggesting a much more permeable blood−brain barrier during ageing than initially anticipated[37]. Candidate genes also showed expression alterations following repeated NSC passaging in an in vitro ageing model further supporting their role in NSC ageing.

Following the in vitro parabiosis model, several components of the IIS pathway, which has been repeatedly linked to NSC regulation and longevity in rodent models, showed associations to human stem cell maintenance[38–42]. *FOXO3A* showed associations to several cellular readouts, in keeping with animal and human studies highlighting it as a key molecule in both longevity and NSC homoeostasis[43,44]. These alterations support its involvement in human NSC ageing and, given its role in oxidative stress and DNA damage response, its decrease in higher-passage cells is indicative of poorer defence mechanisms[45].

Consistent with rodent studies, we confirm a role for the mTOR pathway in NSC regulation and support associations between proposed pathway regulators like GRB10 and PTEN and central components such as mTOR[46–48]. Though GRB10 is known to play an important role in stem cell regulation, its role in NSCs specifically remains unclear. Given its consistently lowered expression following the ageing model, our data support an important function in NSC ageing[49]. Similarly, as PTEN over-expression has been linked to exceptional rodent longevity and to anti-oxidant mechanisms, lowered *PTEN* expression may indicate an age-associated decrease in anti-oxidant response[50–52]. Interestingly, *PTEN* alterations due to passage number occurred only in differentiated treated cells suggesting these are more susceptible to treatment-induced oxidative stress and replicative senescence. Given the different lengths and number of treatment-timepoints in the differentiation and proliferation assays, it remains to be elucidated whether treatment affects NSC progeny more harshly than their undifferentiated counterparts or whether the differentiation assay is more representative of chronic stress due its longer duration and repeated treatment.

The reported alterations in the sirtuin pathway are particularly interesting given its role in mitochondrial regulation and the mounting evidence linking energy metabolism alterations to ageing[1]. Our results align with rodent studies showing NAMPT and UCP2 ablation or deficiency lead to decreased proliferation as both showed a positive correlation to proliferation[53,54]. Though passage number and treatment showed an interesting cumulative effect on *NAMPT* expression, it remains to be elucidated whether this is a negative consequence or a compensatory mechanism. While it is possible that the fluctuation is unrelated to an ageing phenotype, current evidence showing *NAMPT* levels tend to drop in aged stem cells, suggesting this effect is likely a defence mechanism[53,55]. Despite interesting results relating to NAMPT, the rate-limiting enzyme for SIRT1-dependent deacetylation, and UCP2, a protein downstream of SIRT1, *SIRT1* showed no associations to cellular readouts.

Finally, we show several associations between cellular readouts and expression of *ABTB1*, a protein with currently unknown cellular functions besides a probable role in ageing highlighted by Horvath's clock[56]. The results presented here suggest that *ABTB1* expression is involved in NSC regulation and is associated to a

shift towards neuronal differentiation and away from proliferation. To our knowledge, this is the first study linking ABTB1 to NSCs, yet this association is supported by studies showing an interaction between ABTB1 and PTEN, a key regulatory molecule of the mTOR and IIS pathway[57].

Overall, passage number had a far greater effect than chemical treatment. This was surprising given increased passaging being a more likely reflection of chronological, rather than pathological, ageing and previous studies on other cell types employing tBHP and HU as stand-alone stress and ageing models[58,59]. The lack of consistent treatment-effect indicates either a specific resistance of HPCs to these treatments or a lack of coping mechanism. As the cultures contained no microglia and only limited astrocytes, the lack of glial support may have caused inadequate defence mechanisms and extensive cell death. Accordingly, pilot data showed that slightly higher tBHP or HU concentrations (Supplementary Fig. 3A, B) were extremely toxic. Importantly, pilot data confirming individual treatment with tBHP or HU showed no significant effects on cellular (Supplementary Fig. 5A, B) or molecular (Supplementary Fig. 6) data, eliminated the possibility that the combination of treatments masked possible alterations. Though increased passage number may be creating a selection of the most resistant cells, the resulting morphological alterations reminiscent of a senescent phenotype suggest otherwise.

Together, our results show that while the systemic environment can cause significant molecular and cellular changes, these may not be due to chronological age alone. As the systemic environment is strongly affected by lifestyle, and nutrient-sensing pathways are particularly receptive to these alterations, it is likely that the serum donors' varying lifestyles masked age-dependent effects. Accordingly, the alternative ageing model demonstrated age-associated variation in FOXO3A, NAMPT, PTEN and GRB10 which was not detected in the parabiosis model.

We assessed NSC maintenance in ageing models based on the hypothesis that changes in NSC regulation lead to phenotypic alterations such as altered cognitive performance. Based on abundant animal literature and the current lack of techniques to detect in vivo NSC alterations in humans, hippocampal-dependent cognitive performance is often used as a proxy for NSC health[16,60]. Therefore, and as our parabiosis results support a role for lifestyle in NSC maintenance, we tested the association between lifestyle and cognitive performance in a human population to assess whether in vitro associations were reflected in vivo. Our results cannot definitively add to the debate regarding the presence of adult human neurogenesis but, by showing associations of candidate genes to both NSC maintenance and cognitive performance, they are supportive of a link between NSC maintenance and cognition.

We show that several factors affect performance on the hippocampus-dependent task. The correlation between PAL performance and age is supported by existing literature showing an age-dependent decline in episodic memory and by studies using CANTAB PAL both within, and independently of, the TwinsUK cohort[61–63]. Increased education also showed significant effects on cognition but despite recent studies, it remains unclear whether this is due to cognitive reserve or a confounding effect of higher cognitive abilities increasing the likelihood of further education[64–67].

Interestingly, while diet or exercise alone was not associated to cognitive performance, their combination was. The lack of association between calorie intake and cognitive performance contrasts with animal literature where several animal models show decreased calories extend lifespan and improve cognitive abilities[68]. Our results suggest that in a human population this association is confounded by varying exercise levels which are likely to be kept constant under experimental conditions. The suggested inference of our results is that appropriate calorie intake commensurate with energy expenditure is beneficial for cognition, while inappropriately high calorific intake is harmful. However, factors unmeasured in the present study, such as depression or anxiety, could explain this finding. It is important to bear in mind the time difference between dietary, exercise and cognitive assessments in our cohort, which is likely to reduce power to detect associations between lifestyle and cognition but may also be contributing to confounding effects. Despite this, our data suggest the effect of lifestyle and cognition is dependent on the balance between energy intake and expenditure rather than a pre-set association.

Next, we aimed to explain the remaining interindividual variation. Despite several studies reporting associations between FOXO3A polymorphisms and longevity, we report no associations between FOXO3A and cognitive performance[69]. Instead, we report a significant association between rs497849 genotype, a polymorphism near the SIRT1 gene, and PAL performance. Though no other groups have reported direct associations of rs497849 to age-dependent cognition, SIRT1 genotypes have been linked to age and lifestyle-associated traits including cognition, carcinoma, type-2 diabetes and body mass index[70–73]. Importantly, rs497849 is not in the SIRT1 coding region but further downstream near the HERC4 coding region and has unknown consequences on SIRT1 regulation. Despite this, rs497849 has been previously used as a tag-SNP to investigate SIRT1 as it is in strong LD with surrounding SNPs[70,72]. Contrary to other studies, rs497849 was not in LD with any other SNPs possibly due to the limited number of participants causing weaker $r^2$ values which failed to survive the LD threshold.

The association between cognitive performance and rs782431, an SNP near ABTB1, supported a role for this gene in ageing as suggested by our parabiosis data. This SNP was in LD with three other SNPs (rs263021, rs2720262 and rs782455), suggesting any combination of these polymorphisms could be affecting molecular pathways relating to hippocampal cognition. rs782431 is within the MCM2 coding region, which is of particular interest given MCM2's function as DNA replication licensing factor and its use as a marker for slowly cycling NSCs in neurogenesis studies[74,75]. By affecting NSC proliferation, polymorphisms like rs782431 may affect cognitive performance via neurogenesis-dependent mechanisms. Despite this, three of the five most-significantly associated SNPs were in proximity of ABTB1 supporting an important role for ABTB1 in cognition rather than coincidental proximity to the MCM2 gene.

Our study also highlights an interaction between genetics and lifestyle confirming our hypothesis that polymorphisms confer increased susceptibility to certain lifestyles. While other studies have shown positive associations between exercise and memory performance, we showed this was only the case for individuals who were homozygous for the major allele of rs10997817 as the presence of the minor allele appeared to reverse this association[26,27,76–78]. Besides one study finding an association between excessive exercise and poorer mental health, no other has reported negative effects of exercise on cognitive performance[26]. Though assessing the effect of excessive exercise in larger cohorts would provide more information, the higher frequency of the major allele phenotype may explain why most studies find positive effects of exercise, leaving the association between the minor allele, increased exercise and poorer cognition undetected.

We show Mediterranean diet adherence is most beneficial for individuals homozygous for the minor allele of an SNP near GRB10 which is supported by current literature showing MDS is associated to improved cognition[79–81]. Importantly, as this effect was not seen with HEI, our results suggest the MDS's focus on anti-oxidants such as polyphenols is conferring this association.

Given the role of anti-oxidants in neurogenesis, and our in vitro results showing GRB10's association to stem cell maintenance and its expression level alterations following in vitro ageing, our data suggest GRB10 provides the molecular link between diet, nutrient-sensing pathways, NSC ageing and cognitive decline[82–84].

We propose the candidate genes identified here as a starting point for further ageing studies. Overexpression and down-regulation experiments could determine whether the gene expression variations cause the morphological alterations and confirm causal relationships between pathway components. Single-cell analysis could reveal gene-expression signatures across varying morphologies and uncover more genes involved in ageing while determining whether increased passaging is creating a sub-population of resistant super-ager cells.

Though this study's strength is the complementarity of the in vitro and epidemiological results, it has limitations. Owing to the extended timeframe of human ageing, methods such as repeated passaging and chemical treatment were employed to mimic the passing of time and model aspects of ageing rather than replicate exact physiological conditions of multi-cellular or organismal ageing. To assess the role of other cell types in ageing, it would be interesting to compare these results to other relevant human NSC lines, were they to become available. Finally, a larger cohort would enable sufficient power to test for a three-way interaction between diet, exercise and genetics, while longitudinal lifestyle and cognitive data would provide further information on the cognitive decline rate of each participant and on the contribution of lifestyle and genetics to this decline.

In conclusion, through a combination of in vitro and epidemiological approaches, our data support a role for nutrient-sensing pathways in human NSC ageing and cognitive decline. We highlight a novel role for ABTB1 in NSC regulation and cognition and hypothesise a role in nutrient-sensing through the interaction with PTEN. Importantly, GRB10 and the mTOR pathway are highlighted as a likely molecular basis for the association between diet, NSC ageing and hippocampal-dependent cognitive performance. Notably, this study adopted a back-translation approach that used in vitro models to inform epidemiological analysis rather than the reverse. We believe this to be a useful and unbiased approach to investigate healthy ageing while minimising the translational gap which often affects in vitro studies.

## Methods

**Experimental model and subject details.** Cell line: The human hippocampal progenitor cells (HPC) cell line HPC0A07/03A were obtained from ReNeuron and cultured as previously described in refs. [85],[86]. The cells were obtained from 12-week-old foetal female tissue in accordance with UK and USA ethical and legal guidelines and transfected with the *c*-mycER$^{TAM}$ gene construct creating an immortalised cell line that proliferates in the presence of the synthetic drug 4-hydroxy-tamoxifen (4-OHT) and spontaneously differentiates in its absence.

HPC0A07/03A cells were maintained in Dulbecco's modified Eagle's medium supplemented with human albumin solution (0.03 %), Apo-transferrin (100 μg/ml), Putrescine DIHCL (16.2 μg/ml), human recombinant insulin (5 μg/ml), progesterone (60 ng/ml), L-glutamine (2 mM), sodium selenite (40 ng/ml). During proliferation phases, the culture media was additionally supplemented by 4-OHT (100 nM), epidermal growth factor (EGF) (10 ng/ml), basic fibroblast growth factor (bFGF) (20 ng/ml). Cells were cultured in the absence of serum besides when specifically stated as part of in vitro parabiosis experiments. Cells were passaged using Accutase and cultured on Nunclon flasks coated with 20 μg/ml mouse laminin at 37 °C, 5% CO₂ and saturated humidity.

Human subjects: Serum donors. Serum was collected at one timepoint for each participant. Samples from old participants were derived from cognitively healthy control subjects taking part in two ongoing disease biomarker discovery programmes: AddNeuroMed, a multicentre European study[87] and the Maudsley Biomedical Research Centre for Dementia Case Registry at King's Health Partners. All participants scored 27 and above on the Mini Mental State Examination (MMSE) to ensure cognitive health. In both programmes informed consent was obtained per the Declaration of Helsinki (1991). All participants with other neurological or psychiatric disease or illness were excluded. Twenty-two samples

from control subjects were obtained with a mean age of 77.6 years (SD ± 6.43) comprised of 13 females and 9 males. Old participants' age ranged from 53 to 89.

Young participants were recruited through internal advertising within the Institute of Psychiatry, Psychology and Neuroscience between November 2012 and June 2014. Consent and good health were confirmed via a questionnaire as part of the Alzheimer's disease biomarker programmes stated above. Fifteen young control subjects were recruited with a mean age of 27.1 years (SD ± 5.03) comprising of eight females and seven males. Young participants' age ranged from 20 to 32. Protocols and procedures were approved by the appropriate Institutional Ethics Review Board at King's College London.

TwinsUk cohort. Access to phenotypic and genotypic data of the TwinsUK cohort was requested and approved through the TwinsUK Resource Executive Committee (TREC) based at the Department of Twins Research and Genetic Epidemiology at King's College London. Data from 2153 individuals with available cognitive data were received. TwinsUK procedures for sample and data collections have been described previously[63],[88] and were approved by the Guy's and St Thomas' Ethics Committee. TwinsUK data were collected in accordance with the Declaration of Helsinki (1991). TwinsUK is made up of predominantly female white participants of European background. Sample size and demographic information for each trait is summarised in Supplementary Fig. 1.

**Method details.** Serum sample processing: Participants were fasted for 2 h before blood sample collection. Following collection, serum was extracted by centrifugation at 1500 × *g* for 15 min at 4° and stored at −80 °C. Samples were aliquoted to minimise freeze-thaw cycles.

Cell culture details: Cells were seeded in 96-well plates at a density of $1.2 \times 10^4$ cells per well or in six-well plates at a density of $3 \times 10^5$. Proliferation assay: Cells were cultured in proliferation media for 24 h and in proliferation media containing treatment for a further 48 h. Differentiation assay: Following this proliferation phase, the differentiation phase was initiated by washing the cells two times in differentiation media (media without growth factors and 4-OHT) and culturing the cells in differentiation media containing treatment for 7 further days. Treatments included: 1% human serum from young or old persons supplemented with 1% penicillin-streptomycin, 0.01 μM tBHP, 10 μM or 20 μM HU. See Table 1 for reagent details. tBHP was used to induce oxidative stress while HU was employed to induce replicative senescence. HU is an anti-neoplastic drug that represses ribonucleotide reductase and thereby incurs stalled replication forks and eventually double strand breaks in DNA, causing DNA damage and senescence-like changes in proliferating human NSC.

Freeze-thaw cycles of cells were standardised throughout resulting in higher and lower passage numbers both undergoing three freeze-thaw cycles.

Immunostaining: Cells were cultured in 96-well plates for the duration of the proliferation/differentiation assay and fixed in 4% PFA for 20 min at room temperature (RTP). Cells were blocked for 60 min at RTP in blocking solution (phosphate buffered saline (PBS) containing 5% normal donkey serum and 0.3% Triton-X for permeabilization). Primary and secondary antibodies were diluted at 1:500 in PBS except for Mouse anti-Nestin which underwent a 1:1000 dilution. Primary antibody incubation was carried out at 4 °C overnight. Secondary antibody incubation was carried out at RTP for 2 h. Incubation with 300 μM DAPI for 5 min at RTP was carried out for nuclear staining. See Table 1 for reagent details.

Image analysis: All immunostainings were quantified through the semi-automated CellInsight NXT High Content Screening (HCS) platform and Studio software (ThermoScientific) (see Table 1 for software details). This platform relies on light intensity thresholds set by the user, which identify DAPI (wavelength 386) or secondary antibody fluorescence (wavelengths 488 and 555). These thresholds, combined with other parameters based on cell size and shape, can unbiasedly identify cells stained by each antibody and thereby enable semi-automated quantification of immunocytochemical stains. Parameters were kept constant within studies but varied throughout different experimental set-ups.

Candidate gene selection: Candidate genes involved in nutrient-sensing pathways, ageing and stem cell function were selected by literature searches. Genes were selected to span various levels of nutrient-sensing signalling cascades. The selected genes were then assessed for expression within the HPC0A07/03A cell line using existing microarray data (not shown). Those with adequate expression levels were selected as candidate genes for further investigation. The following candidate genes were selected: *mTOR, FOXO3a, SIRT1, MASH1, UCP2, IGF2R, GRB10, PTEN, NAMPT, ETV6 S6K, EIF4e, 4EBP1, IRS2, NRIP* and *ABTB1*.

Gene expression: Cells were cultured in six-well plates for the duration of the proliferation/differentiation assay, lysed in TRIreagent and snap-frozen. Total cellular RNA was extracted by chloroform extraction in Phase lock tubes (Invitrogen) and purification by sodium acetate. TURBO DNA-free Kit was used to digest all DNA. 0.7 g of RNA was used to generate cDNA by incubating it with random hexamers and dNTP mix at 65 °C for 5 min followed by incubation with First Strand buffer, Dithithretiol, RNaseOUT and SuperScript III Reverse Transcriptase at 25 °C for 5 min. Quantitative real-time PCR amplification of cDNA was performed with EVAGreen on Chromo4 Real-Time PCR detector (see Table 1 for reagent details). Relative expression was calculated using the Pfaffl method[89] based on the expression of the three housekeeping genes, *Vimentin, RPLP2* and *ACTG1L*. Gene expression is reported as the relative expression to control condition. In the in vitro parabiosis experiments, the average expression of

**Table 1 Details of key reagents and resources used.**

| Reagent or resource | Source | Identifier |
|---|---|---|
| Antibodies | | |
| Rabbit anti-Ki67 | Abcam | Ab15580 |
| Mouse anti-ki67 | CellSignalling | 9449 |
| Rabbit anti-CC3 | CellSignalling | 9664 |
| Rabbit anti-Sox2 | Abcam | Ab5603 |
| Mouse anti- Nestin | AMD Millipore | Mab5326 |
| Mouse anti-H2a.X | EMD Millipore | 05-636-I |
| Rabbit anti-DCX | Abcam | Ab11267 |
| Mouse anti-Map2 | Abcam | Ab11267 |
| Rabbit anti-NRF2 | Abcam | Ab31163 |
| 555 Donkey Anti-rabbit IgG | Life Technologies | A-31572 |
| 488 Donkey Anti-mouse IgG | Life Technologies | A- 21202 |
| Chemicals, peptides, and recombinant proteins | | |
| Dulbecco's modified Eagle's medium nutrient mixture f-12 ham | Sigma | D6421 |
| Human albumin solution | Zenalb | 20 |
| Apo-transferrin | Sigma | T1147 |
| Putrescine DIHCL | Sigma | P5780 |
| Human recombinant insulin | Sigma | I9278-5ml |
| Progesterone | Sigma | P8783 |
| L-glutamine | Sigma | G7513 |
| Sodium selenite | Sigma | S9133-1MG |
| Epidermal growth factor (EGF) | Peprotech | AF 100-15-500 |
| Basic fibroblast growth factor (bFGF) | Peprotech | EC 100-18B |
| 4-hydroxytamoxifen (4-OHT) | Sigma | H7904 |
| Accutase | Sigma | A1110501 |
| Laminin | Sigma | L2020 |
| Penicillin-streptomycin | Life Technologies | P/S |
| Tert-butyl hydroperoxide (tBHP) | Fisher Scientific | 10703571 |
| Hydroxyurea (HU) | Sigma | H8627 |
| TRIreagent | Sigma | T9424 |
| Random hexamers | Life Technologies | N8080127 |
| dNTP mix | Thermo Scientific | RO191 |
| First Strand buffer | Invitrogen | Invitrogen |
| Dithithretiol | Life Technologies | 18080-044 |
| RNaseOUT | Life Technologies | 10777 |
| SuperScript III Reverse Transcriptase | Invitrogen | 18080093 |
| EvaGreen | Solis BioDyne | 08-24-00008 |
| 100 base-pair DNA ladder | Solis BioDyne | 07-11-00050 |
| 1× DNA Loading Dye | Thermo Fisher Scientific | R0611 |
| DAPI | Sigma | D9542-5mg |
| Critical commercial assays | | |
| TURBO DNA-free™ Kit | Life Technologies | AM1907 |
| Deposited data | | |
| Twins UK cohort data | https://twinsuk.ac.uk/resources-for-researchers/access-our-data/ | NA |
| Experimental models: cell lines | | |
| HPC0A07/03A | ReNeuron Ltd | HPC0A07/03A |
| Oligonucleotides | | |
| mTOR primers | This paper | NA |
| Forward TCTTCCATCAGACCCAGTGA | | |
| Reverse GCTGCCAGCGATCTGAATAA | | |
| GRB10 primers | This paper | NA |
| Forward CACCTGCCTGGCTTCTATTA | | |
| Reverse TGACTGAGGAGCAGAGAAATG | | |
| 4E-Bp1 primers | This paper | NA |
| Forward CGGAAATTCCTGATGGAGTG | | |
| Reverse CCGCTTATCTTCTGGGCTATT | | |
| s6K primers | This paper | NA |
| Forward CATGAGGCGACGAAGGAG | | |
| Reverse GGTCCAGGTCTATGTCAAACA | | |
| eIF4e primers | This paper | NA |
| Forward GAAAAACAAACGGGGAGGAC | | |
| Reverse TCTCCAATAAGGCACAGAAGTG | | |
| IGF2r primers | This paper | NA |
| Forward GAAACAGAGTGGCTGATGGA | | |
| Reverse CTGAGGGCTTTCACTGACTT | | |

**Table 1** (continued)

| Reagent or resource | Source | Identifier |
|---|---|---|
| IRS2 primers<br>Forward CCACCATCGTGAAAGAGTGAA<br>Reverse CAGTGCTGAGCGTCTTCTT | This paper | NA |
| PTEN primers<br>Forward GGTAGCCAGTCAGACAAATTCA<br>Reverse CAACCAGAGTACTACCACCAAAG | This paper | NA |
| ETV6 primers<br>Forward AGGCACCATAATCCCTCCCT<br>Reverse GGGGTCTGCAGCTGTTTAGT | This paper | NA |
| FoxO3a primers<br>Forward GGAGAGCTGAGACCAGGGTA<br>Reverse AGATTCTCGGCTGACCCTCT | This paper | NA |
| Sirt1 primers<br>Forward AGAACCCATGGAGGATGAAAG<br>Reverse TCATCTCCATCAGTCCCAAATC | This paper | NA |
| MASH1 primers<br>Forward GCAGCACACGCGTTATAGTA<br>Reverse ACTCGTTTCTAGAGGGCTAAGA | This paper | NA |
| UCP2 primers<br>Forward CCTCTACAATGGGCTGGTTG<br>Reverse TCAGAGCCCTTGGTGTAGAA | This paper | NA |
| Nrip1 primers<br>Forward GGAGACAGACGAACACTGATATT<br>Reverse GGTCTGTAGCAGTAAGCAGATAG | This paper | NA |
| NAMPT primers<br>Forward CCAGGAAGCCAAAGATGTCTAC<br>Reverse GAAGATGCCCATCATACTTCTCA | This paper | NA |
| ABTB1 primers<br>Forward ACCATGAACCCGTCCTGA<br>Reverse AAGCAGGCATAGTACCTCCA | This paper | NA |
| VIM primers<br>Forward CTTTGCCGTTGAAGCTGCTA<br>Reverse GAAGGTGACGAGCCATTTCC | This paper | NA |
| RPLP2 primers<br>Forward CAGAGGAGAAGAAAGATGAGAAGAA<br>Reverse CTTTATTTGCAGGGGAGCAG | This paper | NA |
| ACTG1L primers<br>Forward GGCTGAGTGTTCTGGGATTT<br>Reverse GGCCAAAGACATCAGCTAAGA | This paper | NA |
| **Software and algorithms** | | |
| QuantStudio 5 Real-Time PCR | ThermoFisher | NA |
| HCS studio software | ThermoScientific | NA |
| Columbus ™ Image Data Storage and Analysis System | Perkin Elmer | NA |
| Prism 6 software (GraphPad Software) | GraphPad | NA |
| R Studio | RStudio Desktop | NA |
| CANTAB PAL | Cambridge Cognition | NA |

the three housekeeping genes across all young serum-treated cells was used as a reference control. In the in vitro ageing experiments, the average expression of the three housekeepers in the control condition (media-only p17) was used as reference control. Housekeeping genes were selected as those with the lowest coefficient of variation between conditions based on microarray data. Primer specificity was checked by agarose gel electrophoresis. Primer sequences are reported in key resources table.

Morphological analysis: Images were taken using the CellInsight platform and analysed using the Columbus™ Image Data Storage and Analysis System version 2.5.0 (Perkin Elmer). DCX- and MAP2-positive cells were highlighted by setting light intensity threshold, cell size and shape parameters as for the CellInsight analysis. Following identification, the linear classifier option was used to train the software to identify each MAP2- or DCX-positive cell as having either an elongated or stunted morphology in an unbiased manner. The machine learning protocol involved a training phase in which a subset of images was used to train the machine to recognise the different morphological properties characterising each subclass. The software was trained on a minimum of five fields per condition. This was followed by a checking phase in which the population selection was checked manually on a subset of fields. Once the parameters were set, the software calculated the percent of cells belonging to each class across the entire plate and identified which morphological properties were used to create the classes.

Cambridge Neuropsychological Test Automated Battery (CANTAB): The digital PAL cognitive task developed by CANTAB was used (Table 1). The task was delivered on a touchscreen and consisted of eight stages of increasing difficulty. Participants are required to recall object locations on the screen. The outcome measure used was the number of errors made by the participant adjusted for trials required and stages completed. The results from the eight-pattern PAL stage were used in this study as these are appropriate for use in cognitively healthy populations. PAL assesses visual memory and learning.

Education, premorbid intelligence and Mini Mental State Examination: Education level was scored from 1 to 8 via a questionnaire instructing participants to indicate the qualification most similar to their highest level of qualification from the following options: No qualification (**1**), NVQ1/SVQ1 (**2**), O-Level/GCSE/NVQ2/SVQ2/Scottish Intermediate (**3**), Scottish higher/NVQ3/city and qulds/Pitman (**4**), A Level, Scottish Advanced Higher (**5**), Higher Vocational training (e.g. Diploma, NVQ4, SVQ4) (**6**), University degree (**7**), Postgraduate degree (e.g. Masters or Ph.D.)/NVQ5/SVQ5 (**8**).

Though some education data were available, performance on NART (errors) was used as a proxy for education and included as a covariate in its place due to greater data availability. NART is widely used as a measure of premorbid intelligence. MMSE data were also collected by TwinsUK, results of 27 and above were used as inclusion criteria in order to exclude subjects with possible dementia.

Lifestyle factors: Average daily calorie intake, HEI and MDS were calculated from food frequency questionnaires (FFQ) collected following the EPIC-Norfolk guidelines. HEI was constructed following the guidelines described by Guenther and colleagues with the exception of the use of the UK based database 'Composition of Foods integrated database' rather than an American based database[35,90].

Briefly, food items are classified into 'U.S. Department of Agriculture' subgroups (e.g. fruit, grains, dairy, etc.) and then categorised into HEI components based on their predominant attributes (e.g. further subdivided into lean and solid fat). Each subgroup has a maximum possible amount of HEI points that can be assigned to it; the FFQ data are converted to a measure of 'nutritional component per 1000 kcal'. Based on this ratio, a certain proportion of the possible HEI points for that subgroup is assigned to the participant. Out of the 12 subgroups, the first 9 are known as adequacy components that aim to quantify the amount of healthy foods consumed, the remaining three (refined grains, sodium and empty calories) were introduced to negatively weigh HEI when excessive amounts of non-healthy foods are consumed. HEI score can range from 0 to 100 points with the maximum achievable score indicating a diet that fulfils all health thresholds. The index was designed so that in a large enough population, 50 would be the mean HEI score and 1 and 99 would be the 1st and 99th percentile. Within the TwinsUK cohort we observe a mean of 60.03, a 1st percentile of 37 and a 99th percentile of 86.

Adherence to MD was evaluated as reported by ref. [91]. A higher score reflects a higher intake of legumes, fruits, nuts, vegetables and olive oil and low intake of saturated lipids.

Physical activity data were collected via the IPAQ[92], which aims at assessing the types of physical activity done by the participants to estimate a total physical activity measure known as MET-minutes per week. Once MET minutes were calculated participants were split into high, moderate and low physical activity following guidelines outlined on the IPAQ website (www.ipaq.ki.se). Data cleaning was also carried out in accordance to IPAQ guidelines; participants with missing answers or with unreasonably high levels of activity were removed and possible human errors when completing the questionnaires were identified. Truncation rules were also applied. Of the participants with both IPAQ and PAL data, all those who declared a form of physical disability were evaluated for inclusion. All those who underwent PAL before the onset of the disability, or underwent PAL less than a year following the disability's onset were excluded as their physical activity levels would not be representative of their physical activity levels at the time of PAL. All those with a disability of unknown duration were also excluded from analysis.

When data were collected at several timepoints, only those closest to the time of PAL were included in the analysis. Out of the 1567 individuals with available PAL and FFQ data, 437 individuals had PAL and FFQ data less than a year apart, 293 had a time interval of 1–3 years between the FFQ and PAL task and the remaining 800 had a time interval greater than 3 years. Out of the 846 participants with eligible PAL and IPAQ data, 473 had undergone the task and IPAQ within a year of one another, 34 within 1–3 years and 339 underwent the task and the IPAQ more than 3 years apart.

Genetic data: Genotype information was obtained for 100 kb up- and downstream of the coding region of each candidate gene. Gene coordinates were selected using the UCSC Genome Reference GRCh37 (assembly date 2009). The genotyping was carried out by TwinsUK using a combination of Illumina arrays (HumanHap300, HumanHap610Q, 1M-Duo and 1.2MDuo 1 M). The data for each array were normalised separately and then pooled. A visual inspection of 100 random shared SNPs across arrays was carried out to rule out any batch effects. Samples were excluded according to the following criteria: a sample call rate <98%, heterozygosity across all SNPs $\geq 2$ SD from the sample mean, evidence of non-European ancestry (assessed by principal component analysis with HapMap3 populations) and possibility of sample identity errors as informed by the identity by descent probabilities. The exclusion criteria included a Hardy–Weinberg $p$ value $< 10^{-6}$ (within unrelated individuals), a minor allele frequency (MAF) of <1% and an SNP call rate of <97% (SNPs with MAF $\geq 5\%$) or <99% (for $1\% \leq$ MAF < 5%). Due to our limited number of participants, MAF was restricted to 5% or more, and a minimum SNP call rate of 95% within our subset was applied. Following this last step of QC there were 4315 eligible SNPs. LD was accounted for by creating tag-SNPs with the Priority Pruner software. This software prunes SNPs by removing all those that are in LD with other SNPs in the dataset. As per convention, an SNP with an $r^2 > 0.8$ and within 250 kb was determined to be in LD. Based on MAF, one of the SNPs in LD is retained and named a 'tag-SNP' while all other SNPs are pruned. Following the creation of tag-SNPs there were 482 eligible tag-SNPs.

**Statistics and reproducibility**. Cellular data: Data were analysed using the GraphPad Prism software. Statistical details of experiments can be found in the respective figure legends. For all $t$ test and correlations, Shapiro–Wilkin test was used to test for normality. As the percentage of KI67-positive cells following in vitro parabiosis did not show a normal distribution, Ki67 data pertaining to that analysis was log-transformed to achieve a normal distribution. In serum experiments, $n$ = number of serum donors. For all other experiments, $n$ = the number of biological replicates, each calculated by averaging three technical replicates. Following significant two-way ANOVAs results, Fisher least significant difference (LSD) was carried out and Bonferroni correction was applied. Correlation, ANOVA and post-hoc results are detailed and reported in the respective results sections.

Epidemiological data: Normality of data was confirmed via visual inspection of frequency and residuals plots. Covariate selection was carried out by testing for association via correlation and $t$ test analysis. To account for relatedness within the cohort, GEE were used instead of linear regressions. The R package 'geepack' was used for all analysis[93]. Each twin pair, irrespective of zygosity, formed a cluster and their relatedness was accounted for within the analysis. An exchangeable correlation structure was selected with the help of Quasi Information Criterion (QIC) and Correlation Information Criterion (CIC). For all analysis involving PAL, age at PAL, NART score and gender were included in the model as covariates. In addition, though normality is not assumed in GEE models, a normal distribution does improve the model's efficiency. The 'PAL errors' variable was log transformed (Log(PAL + 1)) to achieve normality. BH multiple correction was applied at each stage of the analysis. Number of participants, $p$ and beta values are reported in the Results section, further information such as confidence intervals and $q$ values are reported in Supplementary Tables 2–4.

**Reporting summary**. Further information on research design is available in the Nature Research Reporting Summary linked to this article.

## Data availability

Access to the epidemiological data can be requested through the TwinsUK cohort (http://twinsuk.ac.uk/resources-for-researchers/access-our-data/). All other relevant data are in the manuscript or available from the corresponding author upon request. Request for resources and reagents should also be directed to and will be fulfilled by the corresponding author. The raw data used to generate Figs. 2 and 3 can be found in Supplementary Data 5 and 6, respectively.

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

## Acknowledgements

We thank Jane Skinner from the University of East Anglia, for her work converting FFQ data to nutritional information and Ruth C.E. Bowyer from King's College London for the creation of HEI/ MDS from FFQ data. We are also grateful to Genevieve Lachance for facilitating the cohort data transfers. Finally, we thank Curie Kim and Thomas Berger for their help with the revision of this manuscript. Figure 1 was created with Biorender.com. TwinsUK is funded by the Wellcome Trust, Medical Research Council, European Union, the National Institute for Health Research (NIHR)-funded BioResource, Clinical Research Facility and Biomedical Research Centre based at Guy's and St Thomas' NHS Foundation Trust in partnership with King's College London. This work was supported by a grant awarded by the Medical Research Council UK (MR/N030087/1) (S.T.), C.d.L. was supported by the Institute of Psychiatry, Psychology and Neuroscience Ph.D. Prize Studentship Award, T.M. was supported by the Medical Research Council UK (MR/ K500811/1) and the Cohen Charitable Trust. C.J.S. was supported by a grant from the Chronic Disease Research Foundation. P.P. is an Alzheimer's Research UK Senior Research Fellow.

## Author contributions

C.d.L. and S.T. designed the cellular experiments. C.d.L., P.P., C.J.S., R.J.B.D. and S.T. designed the cohort experiments. C.d.L., P.P. and S.T. wrote the paper. C.d.L. and T.M. carried out the in vitro parabiosis experiments. C.d.L. designed and carried out the in vitro ageing model experiments, data collection and analyses. All authors read and revised the manuscript.

## Competing interests

The authors declare no competing interests.
