## [Peer Review File · Communications Biology]

Reviewers' comments:

Reviewer #1 (Remarks to the Author):

I was asked to review the statistical aspects of this article. I find the analyses to be sound and acceptable for publication, and I recommend moving forward with publication based on the statistics covered in the manuscript. However, I also strongly recommend seeking the advice of experts in the scientific content to make sure that substantive claims and broad pieces of the article are also acceptable for publication.

Reviewer #2 (Remarks to the Author):

Lucia et al., Neural stem cells and cognitive aging: Cellular and epidemiological evidence for a role of nutrient-sensing pathways mediated by lifestyle.

Lucia and colleagues present an elegant and comprehensive set of experiments to show contribution of nutrient-sensing pathway components to human stem cell and cognitive aging. It has been suggested that lifestyle impacts the rate of cognitive decline during aging and that nutrient-sensing pathways provide the molecular basis for the association between lifestyle and aging. Since the nutrient-sensing pathways also have been implicated in the maintenance of stem cells that play critical roles in cognition, the authors hypothesized that nutrient-sensing pathways are involved in the interaction of lifestyle, cognition, and stem cell maintenance.

Authors found that expression levels of several key nutrient-sensing genes (FOXO3A, PTEN, IGF2R, mTOR, GRB10, NAMPT, UCP2, and ABTB1) are associated to cellular readouts of stem cell maintenance (markers for stemness, proliferation, differentiation, and apoptosis) in in vitro parabiosis (incubation with human serum) models. Higher passage numbers of stem cells treated with or without tBHP and HU (to better mimic aging), alter the expression levels of FOXO3A, NAMPT and GRB10 after proliferation assay, and FOXO3A, PTEN, and GRB10 after differentiation assay.

Next, authors assessed whether the stem cell maintenance-associated candidate genes identified play a role in cognition in human. Age, gender, and National Adult Reading Scale (NART) were used as covariates. The authors first found the significant associations between SNPs in SIRT1 and ABTB1 with Paired Associates Learning (PAL) performance. Second, authors found that SIRT1 significantly affected the association between physical activity levels and PAL performance, suggesting varying physical activity levels affect PAL performance differently according to genotype. GRB10 significantly affected the association between adherence to Mediterranean diet and PAL score, suggesting that diet is differentially associated with cognitive performance based on genotype. Different approaches converge in highlighting ABTB1 and GRB10 as key genes in the interplay between NSC, aging, lifestyle, and cognition

This study supports a role of nutrient-sensing pathways in human stem cell aging and cognitive decline. The experiments were carefully designed and performed. This is a well-written comprehensive manuscript with exhaustive set of experiments that provide robust results in support of the authors' conclusions. The study has broad public interest and is important. I do not have major concerns. Minor comments are below (can be explained in the revised manuscript). I would recommend acceptance of the manuscript.

Minor

1. SIRT1 was selected as candidate genes for further analysis even though its expression level showed

no correlation to cellular readouts (Page 9). There are more genes that authors found no correlation but also important roles in stem cell maintenance, namely MASH, ETV6, S6K, EIF4E, 4EBP1, IRS2, NRIP (in addition to SIRT1). Among these, why was SIRT1 only included for further analysis? On page 7, (Fig1B) should be (Fig1F).

2. In Figs. 2A and 3B, what do the three dots represent? Please clarify in the figure legend.

3. On page 12, authors mentioned "analysis of mTOR levels following the differentiation assay showed significant variation between groups with 48% of the total variation attributable to passage number." But Fig 2I does not seem to show the difference.

4. On page 17, can "PAL score" be replaced by "PAL errors", as it was used in the figures, to prevent confusion?

Reviewer #3 (Remarks to the Author):

De Lucia et al., follow up on the concepts that environmental/lifestyle and blood borne factors are increasingly important determinants of cognitive aging. Their approach refers to the earlier in vivo parabiosis work by Villeda/Wyss-Coray and others, but uses an in vitro variant with neural stem cells, and an in vitro 'aged' version of them, as readouts to measure changes in gene expression and test links between nutrient-sensing pathways, life style (exercise, diet, etc) and cognitive decline in epidemiological data from human cohorts.

I found this a useful, unbiased and novel approach that will be of interest as it allows researchers to begin to investigate determinants of healthy aging in humans. The link between NSCs and epidemiological data and serum from human cohorts may help to minimize the translational gap often relevant for in vitro studies per se. As such, I applaud the authors for their efforts in developing this extensive paper (69 p), that yields interesting new candidate genes.

Having said that, I found the dataset complex as presented now, and also structure and narrative could be improved for a general audience. I therefore have the following comments and questions to help promote its interesting message;

- 1) Please provide a timeline or chronological/graphic summary of the design and experiments.
- 2) Figures; also when reading only the figures, their story should be easy to follow, which is now often not the case; please clarify/expand the legends further, explain all abbreviations in there and simplify the titles and summarize the main message that the figures display.
- 3) In Fig 1, the values for PTEN and NAMPT do not seem to be distributed homogeneously; how was this dealt with, in particular when compared to the other parameters that often were distributed differently?
- 4) Their in vitro 'aging' model likely has induced a selection of the most resistant cells, and a critic could consider them not (chronologically) old per se. Also, the current difference between young and old is only 9 passages. Why wasn't a bigger difference selected, e.g. 4/5 passages compared to >24?
- 5) Since in their validation experiments of their in vitro aging studies, over 80% of the variation appears to be due to passage number, how is the lack of a major effect of the additional treatments (please make more often clear that tBHP and HU are ment here), interpreted? These seem more relevant factors in real life, whereas passaging per se would more likely reflect chronological ageing. This should be addressed and explained better.

6) Also, while I understand their reasons to 'age' NSCs by increased passaging and additionally exposure them to tBHP and HU, it is quite different from brain aging in general where most neurons do not divide and lack specific stem cell features. In that sense, a group that is only exposed to the tBHP and HU, but without a higher passage number, would have been interesting and relevant. Please discuss.

7) In this respect, in the discussion, it is mentioned that HU was employed to 'induce replicative senescence', but how can the cells then still be passaged? What substrate underlies the senescence in this respect? Is it transient?

8) How is 'healthy' eating in their cohorts scored? This is poorly defined right now. And what does a range between 40 and 80 imply? They refer to a general paper but as important measure, some more specific info would be useful also in their current manuscript.

9) The predictive associations they find for Log(Kcal) and PAL errors, in dependence of IPAQ, are intriguing, yet hard to interpret for me; how to explain this? They may want to present this more clearly and discuss it in more detail.

10) Fixation of cells is said to be in PFA; what percentage and for how long? Why was no pretreatment done before the immunostaining?

11) How was the quantification by machine learning done? It is not mentioned on p.41 and only very briefly on p.42.

12) Given the somewhat limited and often confusing data available so far on the candidate genes identified now, a final, short perspective would be welcome at the end of their discussion on how they could be studied in future experiments to actually confirm their suggested role(s) in cognitive aging.

13) General remark; While the paper does a good job in linking NSC to aging measures, aging of the brain clearly comprises many more modalities and alterations also in other brain regions than 'just' the few stem cells in the dentate gyrus of the hippocampus. It will be important to explain this in their introduction and also mention it as a limitation. In addition, given its clinical and translational relevance, some remarks and references to the recent discussion on the presence of neurogenesis in the human brain, e.g. Kempermann et al., CSC18 and Moreno-Jiminez NatMed 2019, in the introduction and discussion, would be most appropriate.

Minor comments;

- the well-established link between nutrient-sensing pathways and longevity, that is so important for their story, could be stressed some more and better explained/referenced.

- I do not understand their remark on p 27; 'the differentiation assay is a more representative of chronic stress due to its longer duration'. Please explain.

- Given their argument on exercise and their current age group, they'd better replace the Van Praag 2005 citation with another one from the same group on prolonged exercise at middle age; Marlatt/Van Praag 2015. Given their mentioning of cognitive reserve and its possible role in AD, 2 very relevant papers that appeared recently, deserve to be discussed; Lesuis SL et al. Alzheimers Res Ther. 2018 and Stern Y et al., Alzheimers Dement. 2018. Also, when discussing exercise and AD, the recent paper by J Chhatwal in JAMA Neuro 2019 is highly relevant and deserves to be mentioned.

- On P3, the link suggested between AD and NSC lacks appropriate references, while the references mentioned to support the link between NSC to aging and cognition, do not support that argument per se; Yang 15, Romine 15, please rephrase.

- P. 14, 4th line from below; Analysis OF GBR10 expression..

- p. 18; 1st line; ..accurate than therefore, age.. (?)
- p. 20, legend, 2nd line; (C) calorie intake.errors.
- p. 31, 13/4; ..conceivably contributed to confounding.., ..'intake AND expenditure"

Reviewer #4 (Remarks to the Author):

The authors examined expression levels of 16 nutrient-sensing genes in an in vitro ageing model that is developed by the authors. They also examined the association between lifestyle factors and cognitive performance in a human cohort and investigated the role of genetic variation in some genes on cognitive performance. Finally they concluded that the different approaches converge in highlighting ABTB1 and GRB10 as key genes in the interplay between NSC, ageing, lifestyle and cognition. The reviewer has many concerns that should be addressed before the manuscript is accepted for publication. The reviewer would like to suggest re-arrangements of the manuscript. For example, while the authors added TBHP and HU in their in vitro ageing system, they failed to show clear differences. Are those addition essential for this manuscript? The reviewer felt that there are many unnecessary results, which dilute the front line of this manuscript. Although the cohort study demonstrated some potentially interesting information, its relationship to the in vitro study is virtually missed. Critical points were described as below.

Validation of the in vitro ageing model is missed.

Expression levels of the candidate genes were described as relative values. The readers cannot evaluate their expression levels are reasonable or not.

Reasons why the 16 genes are nominated are not explained.

While the in vitro system assessed expression levels of the candidate genes, the in vivo study assessed polymorphisms of which meaning are unknown. Although the authors insist there are some relationship between in vitro and in vivo results, the reviewer found the relationship is not so clear. More careful explanation appears to be necessary.

Almost all discussions missed supportive evidences. For example, the authors suggest that the reason why NAMPT expression is increased in their in vitro ageing system is a defense mechanism. However, it could suggest the in vitro system is improper to assess effects of ageing.

minor points

page 3 lines 5-6 Reference is missed.

page 3 lines 14-16 Reference is missed.

Supplemental tables were not found.

Response to the reviewers for Manuscript COMMSBIO-19-0857-T entitled “Neural stem cells and cognitive aging: Cellular and epidemiological evidence for a role of nutrient-sensing pathways mediated by lifestyle”. By De Lucia et al.

We first would like to thank the editor and the 4 reviewers for their time and diligence in reviewing our manuscript. We have made substantial revisions and have addressed below all the points from the reviewers. We cite the reviewers’ comments in *black/italic* and our responses are in *blue/normal* font. We hope you too will find that the revisions have improved the clarity and quality of our manuscript.

Four reviewers were utilized in this peer review because we wanted at least 1 genetics stats expert to evaluate the statistical analyses:

Referee expertise:

Reviewer 1: Expertise in genomics and statistics

Reviewer 2: Expertise in adult neurogenesis

Reviewer 3: Expertise in stress and adult neurogenesis

Reviewer 4: Expertise in neural stem cells and regeneration

Reviewer #1 (Remarks to the Author):

I was asked to review the statistical aspects of this article. I find the analyses to be sound and acceptable for publication, and I recommend moving forward with publication based on the statistics covered in the manuscript. However, I also strongly recommend seeking the advice of experts in the scientific content to make sure that substantive claims and broad pieces of the article are also acceptable for publication.

Thank you. We are glad to see that the statistical analyses are up to your standards.

Reviewer #2 (Remarks to the Author):

... This study supports a role of nutrient-sensing pathways in human stem cell aging and cognitive decline. The experiments were carefully designed and performed. This is a well-written comprehensive manuscript with exhaustive set of experiments that provide robust results in support of the authors’ conclusions. The study has broad public interest and is important. I do not have major concerns. Minor comments are below (can be explained in the revised manuscript). I would recommend acceptance of the manuscript.

Thank you. We addressed your minor comments below.

1. SIRT1 was selected as candidate genes for further analysis even though its expression level showed no correlation to cellular readouts (Page 9). There are more genes that authors found no correlation but also important roles in stem cell maintenance, namely MASH, ETV6, S6K, EIF4E, 4EBP1, IRS2, NRIP (in addition to SIRT1). Among these, why was SIRT1 only included for further analysis?

Thank you for pointing this out, we have now clarified our reasons behind the inclusion of SIRT1 in the text. We have added an explanation on page 11 (lines 18-19) and expanded this point on page 30 (lines 13-16).

On page 7, (Fig1B) should be (Fig1F).

Apologies. Thank you for pointing out this error. This has been corrected, it is now Fig2F.

2. *In Figs. 2A and 3B, what do the three dots represent? Please clarify in the figure legend.*

Thank you for this comment. We have clarified the meaning of the 3 dots in the legends of figure 3 and 4 (i.e. figure 2 and 3 prior to corrections). We have also clarified and expanded all other legends in the manuscript with the aim to facilitate the understanding of the results they report.

3. *On page 12, authors mentioned “analysis of mTOR levels following the differentiation assay showed significant variation between groups with 48% of the total variation attributable to passage number.” But Fig 2I does not seem to show the difference.*

Thank you for pointing out this needed clarification. The variation in mTOR levels due to passage number did not survive multiple testing correction even though the ANOVA itself was significant. As we only report findings that survive multiple testing in the figures, this finding does not appear significant in figure 3I (i.e. figure 2I prior to corrections). This has now been clarified both in text on page 18 (line 16) and in the figure’s caption.

4. *On page 17, can “PAL score” be replaced by “PAL errors”, as it was used in the figures, to prevent confusion?*

Thank you for highlighting this inconsistency, we have now replaced ‘PAL score’ with ‘PAL errors’ (see page 19 lines 9-11).

Reviewer #3 (Remarks to the Author):

... I found this a useful, unbiased and novel approach that will be of interest as it allows researchers to begin to investigate determinants of healthy aging in humans. The link between NSCs and epidemiological data and serum from human cohorts may help to minimize the translational gap often relevant for in vitro studies per se. As such, I applaud the authors for their efforts in developing this extensive paper (69 p), that yields interesting new candidate genes. Having said that, I found the dataset complex as presented now, and also structure and narrative could be improved for a general audience. I therefore have the following comments and questions to help promote its interesting message;

Thank you. We addressed your comments and questions below.

1) *Please provide a timeline or chronological/graphic summary of the design and experiments.*

We thank the reviewer for this helpful suggestion. We have included a new figure (figure 1) showing a graphic summary of the study design. This figure also clarifies the rationale behind the *in vitro* and *in vivo* experiments.

2) *Figures; also when reading only the figures, their story should be easy to follow, which is now often not the case; please clarify/expand the legends further, explain all abbreviations in there and simplify the titles and summarize the main message that the figures display.*

Thank you for pointing out that the captions required further clarifications. We have now improved all existing figure legends and written a detailed caption for the new figure we have added.

3) *In Fig 1, the values for PTEN and NAMPT do not seem to be distributed homogeneously;*

how was this dealt with, in particular when compared to the other parameters that often were distributed differently?

Thank you for pointing this out. Normality was tested using the Shapiro-Wilks normality test. All data sets that did not show a normal distribution were log transformed, this ensured a normal distribution (as demonstrated by additional Shapiro-Wilks tests). We have now added this information in the figure caption as well. All data sets from figure 2 (i.e. figure 1 prior to corrections) containing Ki67 data underwent this normalization as described in the methods section. When double checking and retesting the data we provided, we noticed we had submitted an old version of the graphs which had incorrect and unlogged axis labels. We have now replaced graphs H, I and J with graphs showing the correct axes.

4) Their in vitro 'aging' model likely has induced a selection of the most resistant cells, and a critic could consider them not (chronologically) old per se. Also, the current difference between young and old is only 9 passages. Why wasn't a bigger difference selected, e.g. 4/5 passages compared to >24?

This is an interesting point. We have therefore added a brief discussion on the possible presence of resistant cells on page 32 (lines 5-9).

We have also clarified the passage number choice on page 12 (lines 7-12) and added the pilot work as a new supplementary Figure (S4).

5) Since in their validation experiments of their in vitro aging studies, over 80% of the variation appears to be due to passage number, how is the lack of a major effect of the additional treatments (please make more often clear that tBHP and HU are meant here), interpreted? These seem more relevant factors in real life, whereas passaging per se would more likely reflect chronological ageing. This should be addressed and explained better.

Thank you for pointing out this omission, we have now clarified our interpretations of the results on page 31 (lines 13-21) and page 32 (lines 1-4). As now explained in the text, we believe the lack of major effects of the pharmacological treatments is due to the relatively low concentrations used throughout the experiments. We have now also discussed that these concentrations had to be used due to the extreme toxicity of higher concentrations and discussed the possible reasons behind these results. We have also added clarifications of what we intend as 'treated' and 'treatment' in several more occasions throughout the manuscript.

6) Also, while I understand their reasons to 'age' NSCs by increased passaging and additionally exposure them to tBHP and HU, it is quite different from brain aging in general where most neurons do not divide and lack specific stem cell features. In that sense, a group that is only exposed to the tBHP and HU, but without a higher passage number, would have been interesting and relevant. Please discuss.

Indeed, throughout the paper we have tested the effect of both treatment and passage number to assess which aspect most resembled *in vivo* brain ageing. All experiments and pilots were carried out in both treated and untreated conditions (figure 2 and figure 3 (previously figures 1 and 2) in the main manuscript and supplementary figures S3 and S5 in the supplementary material). Furthermore, we tested cellular markers and gene expression levels in cells treated with only HU, only TBHP and combination treatments (figure S5 and S6) but saw no consistent effect due to individual or combination treatments. In addition, we also carried out pilot immunohistochemistry experiments with higher concentrations of HU, which resulted in extensive toxicity as shown in supplementary figures S3A, S5A/B. We have now referred to

these results more clearly in the text and added a brief discussion on their relevance on page 31 (lines 13-21) and page 32 (lines 1-4).

7) *In this respect, in the discussion, it is mentioned that HU was employed to 'induce replicative senescence', but how can the cells then still be passaged? What substrate underlies the senescence in this respect? Is it transient?*

Further information on HU and a relevant reference has been added in the methods section on page 44 (lines 16-19). HU causes DNA damage which can lead to double stranded breaks which, in turn, cause cellular stress and trigger the endogenous repair mechanisms. Though the changes induced by HU are not transient (also due to the treatment being over a prolonged period of time), the cell's repair mechanisms are able to fix some of the damage HU induces.

8) *How is 'healthy' eating in their cohorts scored? This is poorly defined right now. And what does a range between 40 and 80 imply? They refer to a general paper but as important measure, some more specific info would be useful also in their current manuscript.*

We agree that more information is necessary for a broad readership. The Healthy Eating Index (HEI) was developed by the United States Department of Agriculture Center for Nutrition Policy and Promotion, and over 700 peer-reviewed scientific papers and federal reports using the HEI have been published since 2008 when the HEI-2005 was released.

A more detailed description of how HEI is calculated and of the meaning of the HEI scores was added in the methods section. We also added a brief description of the MDS. (Page 49 lines 7-23).

9) *The predictive associations they find for Log(Kcal) and PAL errors, in dependence of IPAQ, are intriguing, yet hard to interpret for me; how to explain this? They may want to present this more clearly and discuss it in more detail.*

Thank you for pointing out this needs clarification. We have added a paragraph in the results section (page 21 lines 18-22) to further clarify the results relating to the dependence of the association between Kcal and PAL errors on IPAQ.

10) *Fixation of cells is said to be in PFA; what percentage and for how long? Why was no pretreatment done before the immunostaining?*

Details on the fixation methods were added (page 45 lines 5 and 7). No pretreatment of these cells [grown as monolayer] was necessary besides permeabilization by triton x during blocking (clarified in text).

11) *How was the quantification by machine learning done? It is not mentioned on p.41 and only very briefly on p.42.*

Thank you for highlighting this, further detail on the machine learning process was added in the methods section (page 47 lines 14-20).

12) *Given the somewhat limited and often confusing data available so far on the candidate genes identified now, a final, short perspective would be welcome at the end of their discussion on how they could be studied in future experiments to actually confirm their suggested role(s) in cognitive aging.*

Thank you. This is a very good suggestion. A paragraph has now been added at the start of the newly titled section 'Limitations and future directions'. In this paragraph -on page 38 (lines 2-

11)- we discuss possible future avenues to expand and confirm the findings reported in this manuscript relating to the role of the candidate genes in cognitive ageing.

13) General remark; While the paper does a good job in linking NSC to aging measures, aging of the brain clearly comprises many more modalities and alterations also in other brain regions than 'just' the few stem cells in the dentate gyrus of the hippocampus. It will be important to explain this in their introduction and also mention it as a limitation. In addition, given its clinical and translational relevance, some remarks and references to the recent discussion on the presence of neurogenesis in the human brain, e.g. Kempermann et al., CSC18 and Moreno-Jimenez NatMed 2019, in the introduction and discussion, would be most appropriate.

Thank you for this general remark. The importance of other brain regions in brain ageing has now been mentioned more explicitly on page 3 (lines 13-16) and a new reference discussing how different brain regions are affected by ageing has been added. In addition, the fact that we only focus on a small brain region throughout the manuscript has been discussed as a limitation in the newly titled section ('limitations and future directions') on page 38 (lines 13-14). Furthermore, the currently ongoing discussion on the presence of neurogenesis in the adult human brain is now introduced on page 4 (lines 1-7) and also discussed on page 33 (lines 4-14) alongside the appropriate references mentioned.

Minor comments;

- the well-established link between nutrient-sensing pathways and longevity, that is so important for their story, could be stressed some more and better explained/referenced.

Thank you for this comment. The link between longevity and nutrient-sensing pathways has now been stressed more explicitly on page 5 (lines 7-12 and 17-19) with a new paragraph and several new references backing the importance of each of the 3 pathways the manuscript focuses on.

- I do not understand their remark on p 27; 'the differentiation assay is a more representative of chronic stress due to its longer duration'. Please explain.

We consider that the differentiation assay may be more representative of the chronic aspect of ageing owing to its longer duration and repeated and prolonged treatment. Cells undergoing the proliferation assay alone, are only cultured for 3 days and only undergo 1 treatment of 48 hours, differentiation assay cells instead undergo the same 48-hour treatment as well as a 7-day treatment period during the differentiation stage. This was clarified in text on page 29 (line 22) and page 30 (lines 1-3).

- Given their argument on exercise and their current age group, they'd better replace the Van Praag 2005 citation with another one from the same group on prolonged exercise at middle age; Marlatt/Van Praag 2015. Given their mentioning of cognitive reserve and its possible role in AD, 2 very relevant papers that appeared recently, deserve to be discussed; Lesuis SL et al. Alzheimers Res Ther. 2018 and Stern Y et al., Alzheimers Dement. 2018. Also, when discussing exercise and AD, the recent paper by J Chhatwal in JAMA Neuro 2019 is highly relevant and deserves to be mentioned.

Thank you for the suggestions, we have now rephrased the text and added/replaced the references mentioned above.

- On P3, the link suggested between AD and NSC lacks appropriate references, while the

references mentioned to support the link between NSC to aging and cognition, do not support that argument per se; Yang 15, Romine 15, please rephrase.

The text has now been rephrased for a more accurate description of Yang and Romine's results and new references have been added to support the claim of a link between AD and NSC.

Typos:

- P. 14, 4th line from below; Analysis OF GBR10 expression..

- p. 18; 1st line; ..accurate than therefore, age.. (?)

- p. 20, legend, 2nd line; (C) calorie intake.errors.

- p. 31, 13/4; ..conceivably contributed to confounding.. .. 'intake AND expenditure''

We apologise and we thank the reviewer for taking the time to point out these typos. They have all been addressed.

Reviewer #4 (Remarks to the Author):

1. *The reviewer would like to suggest re-arrangements of the manuscript. For example, while the authors added TBHP and HU in their in vitro ageing system, they failed to show clear differences. Are those addition essential for this manuscript? The reviewer felt that there are many unnecessary results, which dilute the front line of this manuscript. Although the cohort study demonstrated some potentially interesting information, its relationship to the in vitro study is virtually missed.*

Thank you for this comment. We understand that the manuscript is very dense reporting on many aspects of this multidisciplinary study. The pharmacological treatments carried out as part of the in vitro ageing model did not show the hypothesised results. However, separating/removing the treatments results from the passage number results based on hindsight knowledge of non-significant findings would confuse multiple comparison calculations and likely increase the chances of a false positives by effectively lowering the number of multiple comparisons. In addition to this and in agreement with reviewer 3, we believe the absence of alterations due to the pharmacological treatments may make the passage number alterations even more interesting as one would usually predict pharmacological intervention to be more representative of pathological ageing and robust. However, we agree that the more important results were not always highlighted sufficiently, and the negative results not discussed in depth, therefore, we have now added clarifications on these topics on page 31 (lines 13-21) and page 32 (lines 1-9). We have also added clarifications regarding the link between the in vivo and in vitro work on page 33 (lines 1-14).

2. *Validation of the in vitro ageing model is missed. My question is whether the in vitro system is enough reliable as a model of cellular ageing. Without such information, quality of the results obtained from in vitro experiments cannot be evaluated. Already the authors reported a discrepancy in expression of NAMPT in this manuscript. Is it due to compensation, as the author mentioned in the discussion section? It is still possible that the in vitro system developed by the authors is not proper as a model of*

cellular ageing. The reviewer would suggest to compare their in vitro model with established cellular ageing models, such as primary culture of fibroblasts or WI38, and so on.

Thank you for this question. Though there are limitations with the in vitro work, we believe that testing these findings in non-stem cells or even in non-neuronal lineage (e.g. fibroblasts or WI38 as suggested) would introduce more confounders and would not provide a robust validation model. Bearing this in mind, the morphological alterations reminiscent of (in vivo) ageing hippocampal progenitor phenotypes and the alterations in known ageing genes strongly support this model as an *in vitro* ageing model. However, we understand the general point of the reviewer and we have added a paragraph to the limitations section on page 38 (lines 1-16) discussing these issues and further clarified some of the unexpected findings such as NAMPT on page 30 (lines 20-12). We also discuss potential future experiments on page 38 (lines 1-11) that would allow confirmation of the findings in this study.

- 3. I think Expression levels of the candidate genes were described as relative values. The readers cannot evaluate their expression levels are reasonable or not.*

We apologise if this was not clear. qPCR results are reported as the relative expression compared to the control condition. [This is one of the standard practices when reporting qPCR results as it is only a semi-quantitative method. Rather than providing raw values, providing the relative value (calculated using the housekeeper expression levels and the Pfaffl method) provides a way to interpret the fluctuation of gene expression in the other conditions without requiring an absolute value for their expression level].

The origin of the relative values reported throughout the manuscript have now been clarified in both the relevant figure legends on page 11 and page 14/15 and in the methods section on page 47 (lines 1-5).

- 4. Reasons why the 16 genes are nominated are not explained.*

Thank you for highlighting this oversight. We have now included a paragraph in the introduction (page 5 lines 1-12 and 17-21 and page 6 lines 1-2)) explaining the rationale behind the candidate gene choice in addition to the paragraph in the methods section explaining the selection process. We have also emphasised a review previously written by our group which highlights the importance of these genes within their pathways and their link to NSC and ageing.

- 5. While the in vitro system assessed expression levels of the candidate genes, the in vivo study assessed polymorphisms of which meaning are unknown. Although the authors insist there are some relationship between in vitro and in vivo results, the reviewer found the relationship is not so clear. More careful explanation appears to be necessary.*

Thank you for pointing out that this needs further clarification. We have now included a paragraph on page 33 (lines 4-14) explaining further the rationale behind the experimental design and the associations between the *in vivo* and *in vitro* results. We have also included a new figure (Figure 1) which should facilitate the understanding of the relationship between the results.

6. *Almost all discussions missed supportive evidences. For example, the authors suggest that the reason why NAMPT expression is increased in their in vitro ageing system is a defense mechanism. However, it could suggest the in vitro system is improper to assess effects of ageing.*

Thank you for this comment. The possibility that NAMPT expression fluctuation is unrelated to the ageing phenotype has been added in text (page 30 lines (20-21)). On page 38 (lines 13-16) we also added a further limitation discussing the lack of availability of a similar pertinent cell line to carry out validation work. In addition, on page 32 (lines 5-9) and on page 38 (lines 9-11) we now discuss the possibility of a resistant cell population.

minor points:

page 3 lines 5-6 Reference is missed.

page 3 lines 14-16 Reference is missed.

We apologise. Thank you very much for pointing out these omissions and taking the time to highlight them. References have now been added in both places.

Supplemental tables were not found.

We are not sure why. We will resubmit all 4 supplementary tables (one xcel and three CSV files) together with the revised manuscript and figures. We hope they will be accessible.

REVIEWERS' COMMENTS:

Reviewer #1 (Remarks to the Author):

As I said in my first review, this is acceptable for publication based on the statistical analyses performed!

Reviewer #3 (Remarks to the Author):

Thanks for your clear answers and careful modifications. The ms has improved very much. Please check the reference (formats) as often page numbers are lacking now, e.g. in the Kempermann paper. Also the Moreno-Jimenez paper on neurogenesis in ageing and Alzheimer's in Nature Med is not cited, even though from the text, it is inferred this topic is ment. Rather, a different name, Jimenez-Moreno et al is cited, which is I guess not what they mean. Thus, please check all references, and their details, throughout.

Otherwise fine with me; congrats on an impressive and very interesting dataset and paper.

Reviewer #4 (Remarks to the Author):

The authors seriously approached to the comments from referees and the reviewer understood that the manuscript was thoroughly improved. Now the rationale of the manuscript is much clearer than the previous version. However, still the reviewer has some comments on the manuscript. The problem is that the reviewer does not agree with the authors that the in vitro experiments in the manuscript is ageing models. They could be a stress model rather than ageing. The reviewer believes the critical point of this manuscript is investigation of ageing of human cognition and life style. Therefore, the reviewer seeking to understand whether the meaning of in vitro experiment is actually ageing or not.

(1) The reviewer still feel strange with the words "in vitro model". The reviewer thinks that the words "in vitro model" implies an experimental system that mimics something in vivo. The parabiosis model in the manuscript is an experiment examining effects of serum on NSC behaviour in vitro. In vivo, NSCs in the hippocampus is separated from blood with the blood-brain barrier. Blood, as well as serum, cannot reach directly to NSCs in the hippocampus. Of course, some limited contents of serum can path through the BBB, but at least for the reviewer, it appears to be hard to judge the parabiosis model in the manuscript mimics the conditions of NSCs under parabiosis. Accordingly, the authors failed to show a relationship between age and expression of genes of interest. The reviewer would like to suggest to change the name of the experiment. The word "model" appears to be overstatement.

(2) Instead of a relationship between age and expression of genes of interest, the authors demonstrated a relationship between cellular readout and expression of genes of interest. In cellular readouts used in the manuscript, cell density was correlated with 7 out of 9 genes. The authors described that the cellular readouts used in the manuscript is associated with NSC maintenance. In vivo, cell density might not be changed so much. Please explain the reason why cell density is associated with NSC maintenance in the hippocampus.

(3) The second in vitro model was called "in vitro model of NSC ageing" by the authors. Please explain the meaning of NSC ageing. Sometimes people uses "cellular ageing" to describe replicative senescence. However, the reviewer do not know NSCs can be senescence by ageing. As the authors mentioned in the manuscript, neurogenesis in the hippocampus of adult human is not fully elucidated.

However, NSCs in the hippocampus appears to stay as quiescent cells (Aging Cell, 2017 vol. 16(5), pp. 1195-1199). In rodent, the quiescent NSCs can be reversed to proliferating (Cell Stem Cell, 2010 vol. 6(5), pp. 445-456).

(4) "in vitro model of NSC ageing" in the manuscript is combination of pharmacological stress and passage. If passage can be a type of stress for NSCs, the system examined effects of stress, but not ageing, on the protein expression in NSCs in vitro. The reviewer still wonders whether it is proper to call it as in vitro "model" of NSC ageing. Again, the reviewer would like to suggest to change the name of the experiment.

Comments on reply from the authors about qPCR.

Please accept my apology for my bad English, which leads confusion. What the reviewer would like to point out is that "dividing a relative value with a relative value" makes some information disappeared. Of course, results of qPCR is described as the relative value compared to the standard gene, such as house keeping gene. The authors chose the Pfaffl method and microarray, which are more than nice. The point is that the authors showed the qPCR results as the relative compared to the control condition. The reviewer agree to the comments from the authors that it is typical practice recent days. However, genes of which expression levels are similar or comparable to that of gene of interest, at least not too different from, should be selected as standard genes. When results from experimental conditions was divided with that from control condition, it is hard to judge whether the expression levels of standard genes are proper or not. In early phase of qPCR, such information was carefully treated. Nonetheless, the reviewer understood that the authors treated those results properly.

Response to the reviewers for Manuscript COMMSBIO-19-0857B entitled “Lifestyle mediates the role of nutrient-sensing pathways in cognitive aging: Cellular and epidemiological evidence”. By De Lucia et al.

We first would like to thank the editor and the reviewers for their time and diligence in reviewing our manuscript. We have addressed the remaining concerns of the editor and reviewers. We cite the reviewers’ comments in *black/italic* and our responses are in *blue/normal* font. We hope you too will find that the revisions have improved the clarity and quality of our manuscript.

REVIEWERS' COMMENTS:

Reviewer #1 (Remarks to the Author):

As I said in my first review, this is acceptable for publication based on the statistical analyses performed!

Thank you. We are glad to see that the statistical analyses are up to your standards.

Reviewer #3 (Remarks to the Author):

Thanks for your clear answers and careful modifications. The ms has improved very much.

Thank you.

Please check the reference (formats) as often page numbers are lacking now, e.g. in the Kempermann paper. Also the Moreno-Jimenez paper on neurogenesis in ageing and Alzheimer's in Nature Med is not cited, even though from the text, it is inferred this topic is ment. Rather, a different name, Jimenez-

Moreno et al is cited, which is I guess not what they mean. Thus, please check all references, and their details, throughout.

Thank you for highlighting this, the reference formats have been checked and edited.

Otherwise fine with me; congrats on an impressive and very interesting dataset and paper.

Thank you.

Reviewer #4 (Remarks to the Author):

The authors seriously approached to the comments from referees and the reviewer understood that the manuscript was throughly improved. Now the rationale of the manuscript is much clearer than the previous version.

Thank you.

However, still the reviewer has some comments on the manuscript. The problem is that the reviewer does not agree with the authors that the *in vitro* experiments in the manuscript is ageing models. They could be a stress model rather than ageing. The reviewer believes the critical point of this manuscript is investigation of ageing of human cognition and life style. Therefore, the reviewer seeking to understand whether the meaning of *in vitro* experiment is actually ageing or not.

(1) The reviewer still feel strange with the words “*in vitro* model”. The reviewer thinks that the words “*in vitro* model” implies an experimental system that mimics something *in vivo*. The parabiosis model in the manuscript is an experiment examining effects of serum on NSC behaviour *in vitro*. *In vivo*, NSCs in the hippocampus is separated from blood with the blood-brain barrier. Blood, as well as serum, cannot reach directly to NSCs in the hippocampus. Of course, some limited contents of serum can path through the BBB, but at least for the reviewer, it appears to be hard to judge the parabiosis model in the manuscript mimics the conditions of NSCs under parabiosis. Accordingly, the authors failed to show a relationship between age and expression of genes of interest. The reviewer would like to suggest to change the name of the experiment. The word “model” appears to be overstatement.

This has now been addressed to some extent in the text in line with the editor’s comments and suggestions. See page 12, 28 and 38.

We have also added reference regarding the blood brain barrier and accompanying text on page 28.

(2) Instead of a relationship between age and expression of genes of interest, the authors demonstrated a relationship between cellular readout and expression of genes of interest. In cellular readouts used in the manuscript, cell density was correlated with 7 out of 9 genes. The authors described that the cellular readouts used in the manuscript is associated with NSC maintenance. *In vivo*, cell density might not be changed so much. Please explain the reason why cell density is associated with NSC maintenance in the hippocampus.

We now discuss how cell density may reflect the health of the cells in more detail in the text (See page 8).

(3) The second *in vitro* model was called “*in vitro* model of NSC ageing” by the authors. Please explain the meaning of NSC ageing. Sometimes people uses “cellular ageing” to describe replicative senescence. However, the reviewer do not know NSCs can be senescence by ageing. As the authors mentioned in the manuscript, neurogenesis in the hippocampus of adult human is not fully elucidated. However, NSCs in the hippocampus appears to stay as quiescent cells (Aging Cell, 2017 vol. 16(5), pp. 1195-1199). In rodent, the quiescent NSCs can be reversed to proliferating (Cell Stem Cell, 2010 vol. 6(5), pp. 445-456).

(4) “*in vitro* model of NSC ageing” in the manuscript is combination of pharmacological stress and passage. If passage can be a type of stress for NSCs, the system examined effects of stress, but not ageing, on the protein expression in NSCs *in vitro*. The reviewer still wonders whether it is proper to call it as *in vitro* “model” of NSC ageing. Again, the reviewer would like to suggest to change the name of the experiment.

We have adopted the editor’s suggestions to resolve this terminology conflict and this has now been addressed on page 12, 28 and 38.

Comments on reply from the authors about qPCR.

Please accept my apology for my bad English, which leads confusion. What the reviewer would like to point out is that "dividing a relative value with a relative value" makes some information disappeared. Of course, results of qPCR is described as the relative value compared to the standard gene, such as house keeping gene. The authors chose the Pfaffl method and microarray, which are more than nice. The point is that the authors showed the qPCR results as the relative compared to the control condition. The reviewer agree to the comments from the authors that it is typical practice recent days. However, genes of which expression levels are similar or comparable to that of gene of interest, at least not too different from, should be selected as standard genes. When results from experimental conditions was divided with that from control condition, it is hard to judge whether the expression levels of standard genes are proper or not. In early phase of qPCR, such information was carefully treated. Nonetheless, the reviewer understood that the authors treated those results properly.

Thank you. We understand this was a misunderstanding and this is now resolved.